



Atmospheric
Measurement
Techniques

# A study of polarimetric error induced by satellite motion: application to the 3MI and similar sensors

**Souichiro Hioki, Jérôme Riedi, and Mohamed S. Djellali**

Univ. Lille, CNRS, UMR 8518 – LOA – Laboratoire d'Optique Atmosphérique, Lille, 59000, France

**Correspondence:** Souichiro Hioki (souichiro.hioki@univ-lille.fr)

**Abstract.** This study investigates the magnitude of the error introduced by the co-registration and interpolation in computing Stokes vector elements from observations by the Multi-viewing, Multi-channel, Multi-polarisation Imager (3MI). The Stokes parameter derivation from the 3MI measurements requires the syntheses of three wide-field-of-view images taken by the instrument at 0.25 s interval with polarizers at different angles. Even though the synthesis of spatially or temporally inhomogeneous data is inevitable for a number of polarimetric instruments, it is particularly challenging for 3MI because of the instrument design, which prioritizes the stability during a long life cycle and enables the wide-field-of-view and multiwavelength capabilities. This study therefore focuses on 3MI's motion-induced error brought in by the co-registration and interpolation that are necessary for the synthesis of three images. The 2-D polarimetric measurements from the Second-generation Global Imager (SGLI) are weighted and averaged to produce two proxy datasets of the 3MI measurements, with and without considering the effect of the satellite motion along the orbit. The comparison of these two datasets shows that the motion-induced error is not symmetric about zero and not negligible when the intensity variability of the observed scene is large. The results are analyzed in five categories of pixels: (1) cloud over water, (2) clear sky over water, (3) coastlines, (4) cloud over land, and (5) clear sky over land. The most spread distribution of normalized polarized radiance ($L_p$) difference is in the cloud-over-water class, and the most spread distribution of degree of linear polarization (DOLP) difference is in the clear-sky-over-water class. The 5th to 95th percentile ranges of $L_p$ difference for each class are (1) $[-0.0051, 0.012]$, (2) $[-0.0040, 0.0088]$, (3) $[-0.0033, 0.012]$, (4) $[-0.0033, 0.0062]$, and (5) $[-0.0023, 0.0032]$. The same percentile range of DOLP difference for each class are (1) $[-0.023, 0.060]$, (2) $[-0.043, 0.093]$, (3) $[-0.019, 0.082]$, (4) $[-0.0075, 0.014]$, and (5) $[-0.011, 0.016]$. The medians of the $L_p$ difference are (1) 0.00035, (2) 0.000049, (3) 0.00031, (4) 0.000089, and (5) 0.000037, whereas the medians of the DOLP difference are (1) 0.0014, (2) 0.0015, (3) 0.0025, (4) 0.00027, and (5) 0.00014. A model using Monte Carlo simulation confirms that the magnitude of these errors over clouds are closely related to the spatial correlation in the horizontal cloud structure. For the cloud-over-water category, it is shown that the error model developed in this study can statistically simulate the magnitude and trends of the 3MI's motion-induced error estimated from SGLI data. The obtained statistics and the simulation technique can be utilized to provide pixel-level quality information for 3MI Level 1B products. In addition, the simulation method can be applied to the past, current, and future spaceborne instruments with a similar design.

## 1 Introduction

The Multi-viewing, Multi-channel, Multi-polarisation Imager (3MI) is a planned spaceborne sensor on the MetOp Second Generation-A (MetOp-SG-A) satellite platform. The sensor consists of two wide-field-of-view cameras with narrowband wavelength filters, inheriting the legacy of the POLarization and Directionality of the Earth's Reflectance (POLDER) sensor. The rotating wheel carries 31 filters that enable polarimetric measurements at nine wavelengths and non-polarimetric measurements at three wavelengths (Fougnie et al., 2018). The spatial resolution at nadir is 4 km, and the instantaneous swath is 2200 km. The MetOp-SG-A se-

ries expects the launch of three identical platforms with 7-year intervals, providing continuous and homogeneous monitoring of the Earth's weather and climate for 21 years. The 3MI sensors on these platforms are anticipated to perform multi-viewing and multi-channel polarimetric observation at unparalleled spatial and temporal scales.

As increasingly advanced retrieval techniques are used to extract atmospheric composition parameters from multispectral and polarimetric observations (Knobelspiesse et al., 2012; Fu and Hasekamp, 2018; Dubovik et al., 2019), the knowledge and reduction of uncertainties associated with polarimetric measurements become more and more critical. In particular for all techniques relying on optimal estimation, the correct understanding of observational uncertainties, as discussed in detail by Povey and Grainger (2015), is pivotal to obtain meaningful and successful retrieval. The best-described uncertainty in the Level 1B product is the radiometric uncertainty. Table 1 summarizes the radiometric requirements of the 3MI sensor according to Fougnie et al. (2018). With no onboard calibration, the 3MI sensor needs to rely on in-flight vicarious calibration techniques to monitor and assess the instrument radiometric performance. Among vicarious calibration techniques that are developed for the PARASOL sensor (Fougnie, 2016) and new techniques (e.g., Djellali et al., 2019), some are useful for an independent absolute calibration and others are more suited to perform inter-band or inter-sensor cross-calibration. For example, the Rayleigh scattering method and the new sunglint method can be used to monitor the absolute calibration coefficient, whereas existing sunglint method is designed for inter-band calibration. The temporal evolution of calibration coefficients could be monitored by the calibration over Antarctica, and inter-pixel calibration could be performed over deep convective clouds. The synergies with other sensors of the MetOp-SG-A platform will also be beneficial (Fougnie et al., 2018), e.g., through a cross-calibration over a relatively invariant target over the African and Arabian deserts.

The radiometric performance of the instrument, however, is not the only factor driving the overall measurement uncertainties, especially for polarimetric observations where the useful quantities (Stokes parameters) are not directly measured but derived, e.g., from a set of different radiances. Therefore, the polarimetric performance of the 3MI instrument will depend on both the radiometric accuracy and the process used to derive the Stokes parameters. This paper investigates in particular the uncertainty induced by this derivation process in order to provide an instantaneous, realistic, and quantitative estimate of polarimetric error at pixel level for the 3MI instrument or similar sensors.

The 3MI instrument derives the first three Stokes parameters ($I$, $Q$, and $U$) by synthesizing three wide-field-of-view images that are taken sequentially at a 0.25 s interval. Each of the three images is acquired with the linear polarizer oriented in different directions with the polarizing axis being

**Table 1.** Specification of the radiometric accuracy for the 3MI instrument (summarized from Fougnie et al., 2018).

| Subject | Requirements (better than) |
|---|---|
| Absolute calibration accuracy | 2 % |
| Inter-band (spectral) calibration consistency | 1 % |
| Lifetime radiometric consistency | 1 % |
| Inter-view calibration consistency | 2 % |
| Pixel-to-pixel relative calibration consistency within any $10 \times 10$ pixel area | 0.1 % |
| Knowledge of instrumental sensitivity to polarization | $10^{-3}$ |
| Signal-to-noise ratio (SNR) at channel reference radiance value | 200 |

−60, 0, and 60° with respect to the direction of the satellite's orbital motion (along-track direction). However, within the 0.25 s interval, the instantaneous field of view (IFOV) shifts by 1.8 km (0.45 pixel) on the ground because of the motion of the satellite. The shifts between the acquired images require interpolation and co-registration to compensate for the satellite's along-track motion before the computation of the Stokes parameters. These non-simultaneous acquisitions therefore introduce a possible source of error in the polarimetric observation by the 3MI sensor due the co-registration and the interpolation required to match the three images.

This kind of error due to the synthesis of spatially or temporally inhomogeneous data is inevitable for polarimetric instruments without a beam splitter, including the 3MI, because it is impossible to measure multiple radiometric quantities along the same line of sight simultaneously and independently. The practical solution is to spatially or temporally change the polarimetric modulation in a way that minimizes the errors while at the same time providing measurements that serve the mission objectives. For the details of the different instrument designs, readers are referred to Dubovik et al. (2019) and references therein. The rotating-filter design of the 3MI and POLDER sensors is suited for the spatially continuous wide-field-of-view measurements, but its asynchronous acquisition may result in a significant polarimetric error, particularly over inhomogeneous scenes.

The magnitude of error due to the interpolation and co-registration is expected to be neither spatially uniform nor symmetric about zero, and it therefore cannot be removed by spatiotemporal averaging. This is because the intensity in the original images and the polarized normalized radiance ($L_p$) are not linearly related. We define the normalized radiance ($L$) and, in analogy, the polarized normalized radiance

as follows:

$$L = \frac{\pi I}{E_0}, \tag{1}$$

$$L_p = \frac{\pi}{E_0} \sqrt{Q_i^2 + U_i^2}, \tag{2}$$

where $I$ is the intensity, $Q_i$ and $U_i$ are the second and third
elements of Stokes vector in terms of intensity, and $E_0$ is the
beam flux of the extraterrestrial solar radiation. The degree
of linear polarization (DOLP) is defined as follows:

$$\text{DOLP} = \frac{L_p}{L}. \tag{3}$$

Assuming three ideal linear polarizers with a perfect align-
ment, the normalized radiance and the polarized normalized
radiance can be computed from the original intensity mea-
surements $(X_{m60}, X_0, X_{p60})$ as follows:

$$L = \frac{2}{3} \frac{\pi}{E_0} \left( X_{m60} + X_0 + X_{p60} \right), \tag{4}$$

$$L_p = \frac{2\sqrt{2}}{3} \frac{\pi}{E_0}$$
$$\sqrt{(X_{m60} - X_0)^2 + (X_0 - X_{p60})^2 + (X_{p60} - X_{m60})^2}, \tag{5}$$

where $X_{m60}$ corresponds to the intensity with the polarizer
aligned $-60°$ off from the along-track direction, $X_0$ to $0°$
off, and $X_{p60}$ to $+60°$ off. Equation (5) demonstrates that the
error in the original images and the normalized polarized ra-
diance are not in linear relation. Rather, the motion-induced
error tends to suppress the polarization for strongly polarized
target and tends to enhance the polarization for weakly po-
larized target. This is because $L_p$ is bounded by $0 < L_p < L$
(i.e., $0 < \text{DOLP} < 1$), and any random error near $L_p = 0$ in-
creases $L_p$, whereas that near $L_p = L$ decreases $L_p$. From
this discussion, we emphasize again that the motion-induced
error likely depends on the observation target, and the spatial
distribution may not be uniform.

As the quality of the retrieval product hinges on the qual-
ity of the radiance product and ancillary information that de-
livers the reliability, the quantification of the error triggered
by the interpolation and co-registration is significantly use-
ful in the quality control of both Level 1B and Level 2 data
products. Some studies have already investigated the impact
of the polarimetric accuracy on the performance of retrieval
algorithms (e.g., Hasekamp et al., 2019), but the quantifica-
tion of the error in Level 2 data products is still challenging
not only because the retrieval error highly depends on the
retrieval algorithm but also because the characterization of
errors in Level 1B data products is still largely limited to the
radiometric uncertainty. By addressing the characteristics of
the motion-induced error, this study therefore adds a new di-
mension to the reliability of the Level 1B and Level 2 data
products.

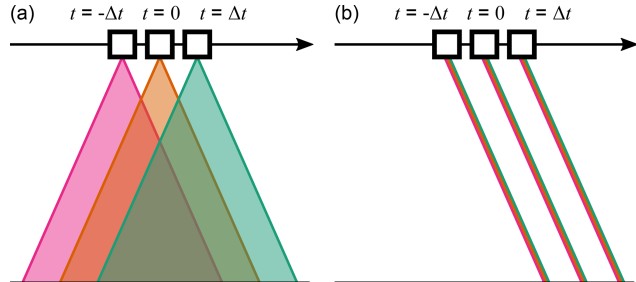

**Figure 1.** The schematic diagram of the field-of-view by the **(a)** 3MI
and **(b)** SGLI. The black arrow shows the motion of the satellite
along the orbit, and the three positions of the satellites along the
track are shown to highlight asynchronous acquisition by the 3MI
sensor.

Section 2 describes the data and methods used in this
study, Sect. 3 describes the results, and Sect. 4 discusses the
simulation of the error statistics. Conclusions are given in
Sect. 5.

## 2 Data and methods

### 2.1 Data

In this study, we produce 3MI proxy polarimetric measure-
ments from the actual high-resolution polarimetric measure-
ments obtained by the Second-generation Global Imager
(SGLI) aboard the Global Change Observation Mission –
Climate (GCOM-C) satellite (Imaoka et al., 2010). The SGLI
sensor provides polarized normalized radiances at 1 km nadir
resolution within a 1150 km wide swath. We selected the
SGLI data as a source of our proxy data, because they pro-
vide the global polarimetry at higher resolution than the 3MI;
it suffers less from the co-registration and interpolation er-
rors, and its orbit and swath are compatible to that of 3MI.

Because of the push-broom design of the SGLI sensor,
SGLI's interpolation error is expected to be negligible com-
pared to that introduced by the 3MI's more complex co-
registration and interpolation. Figure 1 is a schematic dia-
gram showing the difference between the observation geom-
etry of the 3MI and that of the SGLI. Both sensors mea-
sure the first three elements of the Stokes vector through the
measurements of intensity through linear polarizer oriented
at three different directions. The pronounced difference is
that the 3MI sensor acquires the three wide-field-of-view im-
ages asynchronously, while the SGLI sensor acquires three
cross-track profiles of intensity with 170 m offset. Because
the SGLI sensor design suffers less from the shift between
the acquisition of three images, we treat the SGLI measure-
ment as truth and evaluate the magnitude of error introduced
by the interpolation and co-registration by producing 3MI-
proxy data from the SGLI data.

One week of global SGLI Level 1B data near the 2008 September equinox (20–26 September) are used. In this study, we performed our analysis based on the polarimetric data in the Level 1B POLDK product and visible, near infrared, and thermal infrared data in Level 1B VNRDK, VN-RDL, IRSDK, and IRSDL products. The product name conventions are described in the GCOM-C Data Users Handbook, but in short, the first three letters indicate the subsystem of the SGLI instrument ("POL" for polarization, "VNR" for visible and near infrared, and "IRS" for infrared scanning subsystems), the fourth letter "D" indicates the observation mode ("D" for daytime data), and the last letter indicates the resolution ("K" for 1 km resolution, "L" for aggregated 1 km resolution). As the GCOM-C satellite is in the sun-synchronous orbit at 800 km altitude with descending-node local time of 10:30 LT, the data from the SGLI are valuable to simulate measurements from the 3MI that are anticipated also in the sun-synchronous morning orbit at 830 km altitude.

## 2.2 Methods

### 2.2.1 Estimation of the motion-induced error

As the nominal resolution of the 3MI is 4 km and that of the SGLI is 1 km, $4 \times 4$ SGLI pixels are aggregated to produce a 3MI pixel. We repeat the aggregation for every $4 \times 4$ SGLI pixel blocks in three original ("Lt_P2_m60", "Lt_P2_0", and "Lt_P2_p60") radiance datasets in the SGLI Level 1B product. These three datasets correspond to the radiance measured at 0.869 μm with polarizers at three different directions at $-60$, 0, and $60°$ with respect to the along-track axis of the satellite (i.e., $X_{m60}, X_0, X_{p60}$). From the aggregated data, we compute the normalized radiance and polarized normalized radiance by Eqs. (4) and (5). The results are referred to as "reference data" hereafter.

A similar method is applied to produce the 3MI "proxy data" which are compared to the reference data to estimate the magnitude of the error. The proxy data are different from the reference data in that they incorporate the effects of the satellite's motion and interpolation. To simulate the motion of satellite between the acquisition of each image (1.8 km), we compute the contribution of every SGLI pixel to the shifted $4 \times 4$ grid by the following equation:

$$w_0(i) = \frac{1}{16} \int_{i-1}^{i} \Pi_{4,8}(x-s)\, \mathrm{d}x, \tag{6}$$

where $i$ is the index of line, $\Pi_{a,b}(x)$ is a boxcar function that is 1 in the interval $(a,b)$ and 0 otherwise, and $s$ is the amount of shift in SGLI pixel size (i.e., $+1.8$ or $-1.8$ in our case). The results of the calculation is shown in Table 2. The contribution factors are multiplied by the measured radiance in every SGLI pixel to perform the weighted average.

After computing the weighted average, we obtain the shifted, aggregated images of "Lt_P2_p60" and "Lt_P2_m60". As these two images are shifted with respect

to the "Lt_P2_0" image, the unshifting process is performed as a next step by interpolating the intensity at the pixel center location of the "Lt_P2_0" image. The linear interpolation is selected for the simplicity and the locality, but the error estimation described in this subsection as well as the error simulation detailed in Appendix A could be performed with other methods of interpolations if deemed necessary for specific applications. The final contribution factors are computed with the following equation:

$$w(i) = \left(1 - \frac{s}{4}\right) w_0(i) + \frac{s}{4} w_0(i + 4\,\mathrm{sgn}(s)), \tag{7}$$

where $\mathrm{sgn}(s)$ is sign of the shift. The results of the computation are summarized in Table 3. From these unshifted aggregated images, we compute the normalized radiance ($L$) and the normalized polarized radiance ($L_p$) by Eqs. (4) and (5), and we call them the proxy data.

The comparison of the proxy and reference data is performed on a pixel-by-pixel basis. In every pixel, the difference in polarized normalized radiance $\Delta L_p$ and the difference in degree of linear polarization $\Delta \mathrm{DOLP}$ are computed. These differences are attributed to the error induced by the pixel co-registration and the interpolation.

### 2.2.2 Classification of data

To further the analysis of the motion-induced error, we classify pixels into five categories: clouds over water, clear sky over water, clouds over land, clear sky over land, and coastline. The classification is based on the land–water flag in the SGLI Level 1B dataset and the cloud flag algorithm developed for this study. The activity diagram (flowchart) of the cloud flag algorithm is shown in Fig. 2, while individual test conditions are listed in Tables 4 (pixels over water) and 5 (pixels over land). To compute cloud flags in the SGLI Level 1B POLDK product's coordinate, other SGLI L1B products are projected onto the POLDK grid. Once the cloud flag is derived, both land–water flag and cloud flag are extended into the along-track directions by 20 SGLI pixels to cover all SGLI pixels used for the error estimation and to minimize the pixels on boundaries. A pixel is classified as "cloud over water" when it is flagged as "confidently cloudy" and "land cover 0 %", "clear sky over water" when it is flagged as "confidently clear" and "land cover 0 %", "clear sky over land" when it is flagged as "confidently clear" and "land cover 100 %", "cloud over land" when it is flagged as "confidently cloudy" and "land cover 100 %", and "coastline" when land cover is between 5 % and 95 %. Only pixels with glint angle greater than $35°$ are collected for cloud-over-water, clear-sky-over-water, and coastline classes to avoid contamination by sunglint.

In every class of data, the normalized polarized radiance differences ($\Delta L_p$) are regrouped according to the along-track Laplacian that is defined as follows:

$$L_{\mathrm{AT}} = \frac{\pi}{E_0}(2X_0 - X_{-1} - X_{+1}), \tag{8}$$

**Table 2.** Contribution of SGLI pixels to shifted and unshifted averaging grids for a single 3MI pixel. These weights are intended to simulate the along-track motion of the satellite.

| Line | Weights for the unshifted grid | Weights for the shifted grid (+1.8 km) | Weights for the shifted grid (−1.8 km) |
|---|---|---|---|
| 1 | 0 | 0 | 0 |
| 2 | 0 | 0 | 0 |
| 3 | 0 | 0 | 1/20 |
| 4 | 0 | 0 | 1/16 |
| 5 | 1/16 | 0 | 1/16 |
| 6 | 1/16 | 1/80 | 1/16 |
| 7 | 1/16 | 1/16 | 1/80 |
| 8 | 1/16 | 1/16 | 0 |
| 9 | 0 | 1/16 | 0 |
| 10 | 0 | 1/20 | 0 |
| 11 | 0 | 0 | 0 |
| 12 | 0 | 0 | 0 |

**Table 3.** Contribution of SGLI pixels to shifted and unshifted averaging grids for a single 3MI pixel. These weights include the effects of the satellite's along-track motion and the interpolation.

| Line | Weights for the unshifted grid | Weights for the shifted grid (+1.8 km) | Weights for the shifted grid (−1.8 km) |
|---|---|---|---|
| 1 | 0 | 0 | 0 |
| 2 | 0 | 9/1600 | 0 |
| 3 | 0 | 9/320 | 11/400 |
| 4 | 0 | 9/320 | 11/320 |
| 5 | 1/16 | 9/320 | 11/320 |
| 6 | 1/16 | 47/1600 | 11/320 |
| 7 | 1/16 | 11/320 | 47/1600 |
| 8 | 1/16 | 11/320 | 9/320 |
| 9 | 0 | 11/320 | 9/320 |
| 10 | 0 | 11/400 | 9/320 |
| 11 | 0 | 0 | 9/1600 |
| 12 | 0 | 0 | 0 |

where $X_{-1}$ is the $X_0$ of the adjacent pixel in the negative along-track direction, and $X_{+1}$ is the $X_0$ of the adjacent pixel in the positive along-track direction. The along-track Laplacian is a measure of the nonlinearity of the local intensity change in the along-track direction. We select this metric because the linear interpolation in Eq. (7) removes the linear error term, and the highest remaining term is characterized by the Laplacian. In other words, we estimate that the major source of the error is the (nonlinear) spatial inhomogeneity of the total radiance that pass through the linear interpolation performed in the co-registration.

The DOLP differences ($\Delta$DOLP) are regrouped according to the along-track Laplacian divided by $L$, i.e., $L_{AT}/L$. This is because we found from the preliminary study that the relation between $\Delta L_p$ and $L_{AT}$ does not strongly depend on the value of $L$. As the DOLP is defined by Eq. (3), utilizing $L_{AT}/L$ as a regrouping parameter is deemed a reasonable choice.

# 3 Results

## 3.1 Estimation of the error

The difference in DOLP between the proxy and reference data is shown in Fig. 3a with corresponding visible composite in Fig. 3b. The figure covers the western French coast, English Channel, and southern Britain. Figure 3a shows that the DOLP difference can reach more than 0.2 along the coastlines, particularly in a part where the coastline runs in the cross-track direction. There is little negative DOLP difference near positive values, implying that the distribution of the DOLP difference is not symmetric about zero. Along the edges of thin clouds, e.g., over the Atlantic in the western part of Fig. 3, positive and negative DOLP differences are mixed. Note that similar clouds over the ocean and land give different magnitudes of DOLP differences, presumably because of the background normalized radiance ($L$). As the DOLP is

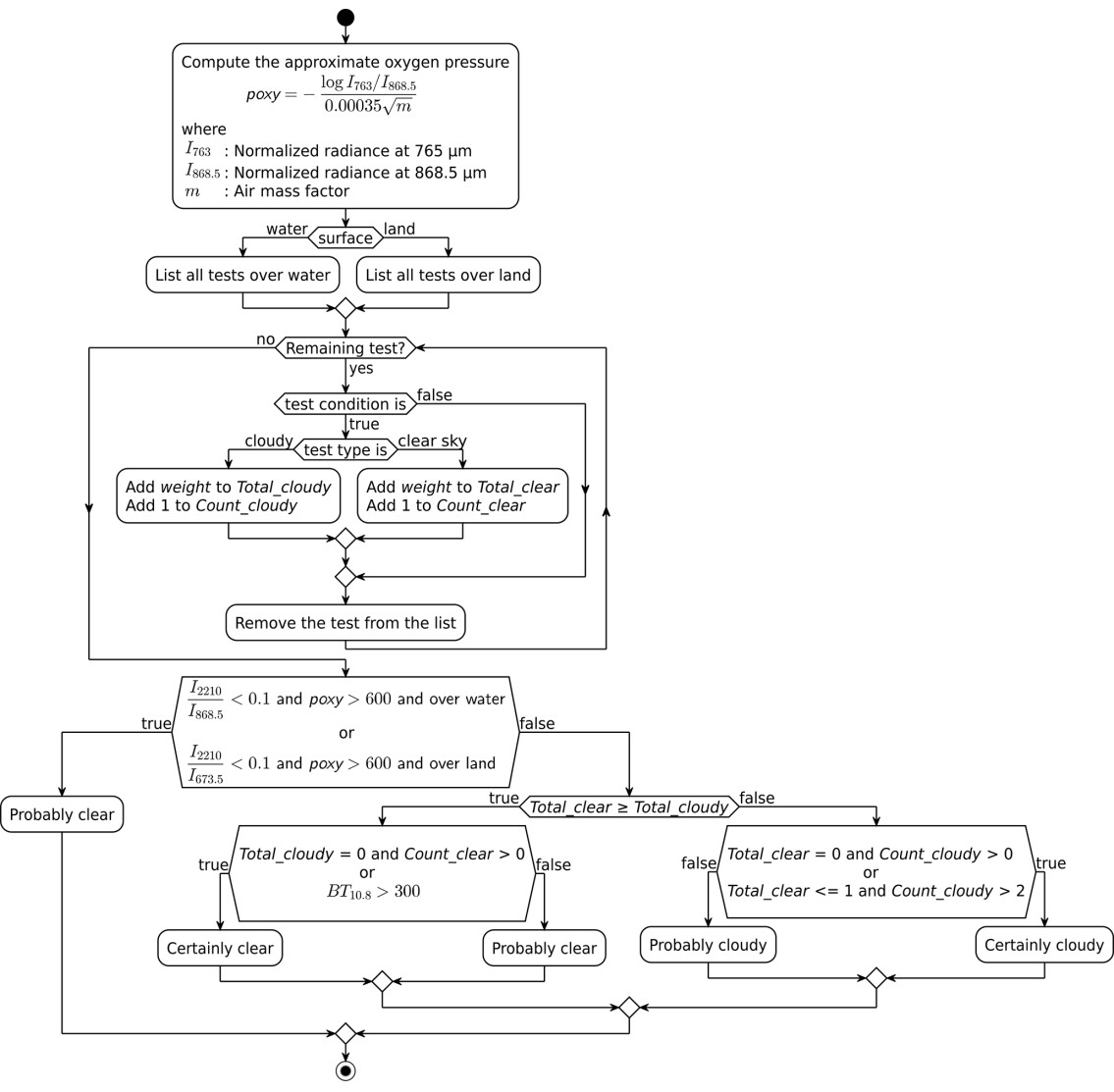

**Figure 2.** The activity diagram (flowchart) of the cloud flag algorithm.

**Table 4.** Tests for the cloud detection algorithm over water. $I_{XXX}$ indicates the normalized radiance at wavelength $XXX$ nm, $BT_{XX}$ indicates the brightness temperature at $XX$ μm, and $\sigma_{XXX}$ the standard deviation of $I_{XXX}$ in the concentric box of $3 \times 3$ pixels.

| Test condition | Test type | Weight |
|---|---|---|
| $(I_{1630}/I_{868.5} < 1.2)$ and $(BT_{10.8} < 288\,\mathrm{K})$ and $(I_{673.5} > 0.2)$ | Cloudy | 1 |
| $I_{673.5} > 0.45$ | Cloudy | 10 |
| $(I_{673.5} > 0.35)$ and $(\sigma_{673.5} > 0.01)$ | Cloudy | 10 |
| $(I_{673.5} > 0.15)$ and $(1 < I_{673.5}/I_{868.5} < 1.1)$ and $(BT_{10.8} < 295\,\mathrm{K})$ | Cloudy | 10 |
| $(I_{673.5} > 0.2)$ and $(\sigma_{673.5} > 0.005)$ and $(I_{673.5}/I_{1630} < 1.4)$ | Cloudy | 100 |
| $(BT_{10.8} - BT_{12} < -1\,\mathrm{K})$ and $(BT_{10.8} < 300\,\mathrm{K})$ | Cloudy | 1000 |
| $(I_{1630}/I_{868.5} > 1.3)$ and $(BT_{10.8} > 300\,\mathrm{K})$ | Clear sky | 1 |
| $(I_{673.5}/I_{868.5} < 0.7)$ | Clear sky | 10 |
| $(\sigma_{673.5} < 0.1)$ and $(0.2 < I_{673.5} < 0.5)$ and $(1.5 < I_{1630}/I_{673.5} < 10)$ | Clear sky | 100 |
| $I_{1630}/I_{673.5} > 2.2$ | Clear sky | 100 |
| $(BT_{10.8} - BT_{12})$ and $(BT_{10.8} > 300\,\mathrm{K})$ | Clear sky | 1000 |
| $\left\lvert \dfrac{I_{868.5} - I_{673.5}}{I_{868.5} - I_{1630}} \right\rvert > 2$ | Clear sky | 1000 |

divided by $L$, errors over dark pixels (e.g., ocean and water near coastlines) tend to be pronounced. For the same reasons, the magnitude of the error is smaller over thick clouds covering the eastern end of English Channel than along coastlines. Figure 3 demonstrates that the magnitude of motion-induced error is not negligible over the scene where intensity variation is large.

It is not only over coastlines that the distribution of error is asymmetric about zero. In Figs. 4 and 5, we show histograms of degree of linear polarization (DOLP) and polarized normalized radiance ($L_p$) differences for five different classes of pixels defined in Sect. 2.2.2: cloud over water, clear sky over water, cloud over land, clear sky over land, and coastline at wavelength $\lambda = 0.869\,\mu m$, in addition to the clear sky over land at wavelength $\lambda = 0.674\,\mu m$. Figures 4 and 5 show that the distributions of the proxy–reference differences are not symmetric about zero and rather skewed to the right (having fat tail in the right end of the histogram) in all classes of data. The tail of DOLP difference histogram is particularly fat for cloud-over-water and coastlines classes, where sharp reflectance gaps near coastal water or cloud edges induce strong polarization artifacts.

The asymmetry can be confirmed from the 5th to 95th percentile range and 25th to 75th percentile range (interquartile range) of the estimated DOLP and $L_p$ error as presented in Table 6. The median of distribution shown in Table 7 also indicates that the distribution is asymmetric not only about zero but also about the median. The clear-sky-over-water class has the most spread distribution of DOLP differences, and the 5th-95th percental range is $[-0.043, 0.093]$. On the other hand, the cloud-over-water class has the most spread distribution of $L_p$ differences, and the 5th to 95th percentile range is $[-0.0051, 0.012]$. The asymmetry of the distribution implies that the error cannot be completely canceled out by computing the spatial average.

Figure 6 shows the proxy–reference DOLP differences ($\Delta$DOLP) as a function of the along-track Laplacian ($L_{AT}$). The red curve indicates the median of $\Delta$DOLP values in each bin of $L_{AT}$, and the gray shadow the interquartile range of $\Delta$DOLP values. As the $L_{AT}$ increases, the median of the error as well as the spread increase. However, an evident difference exists between the clear sky over land at $\lambda = 0.869$ and $\lambda = 0.674$. The median of $\lambda = 0.674$ shows a slight increase as a function of $L_{AP}$ but remains small (within 0.01). At $\lambda = 0.869$, for cloud-over-land, cloud-over-water, and clear-sky-over-water classes, the spreads of the distributions saturate when $L_{AT}/L$ increases, while for coastlines the spread of the distribution does not saturate. In the clear-sky-over-land class, the spread reaches a peak at $L_{AT}/L = 0.15$. The strong dependence of medians to the along-track Laplacian imply that the magnitude of the error is predictable from the image of $X_0$ (intensity with polarizer at $0°$ off the along-track direction), which is used to compute the $L_{AT}$ and observationally available.

The distribution of polarized normalized radiance difference ($\Delta L_p$) is also shown to be a function of the along-track Laplacian ($L_{AT}$), and Fig. 7 show the increase in median with increasing $L_{AT}$. The spectral difference for clear-sky-over-land class still exists, but less pronounced than that for the DOLP. The median of $\lambda = 0.674\,\mu m$ is closer to 0 than the median of $\lambda = 0.869$ in the range $[0, 0.02]$ where a majority of pixel exists. It is plausible that the low land inhomogeneity at $\lambda = 0.674\,\mu m$ results in lower $L_p$ and DOLP errors. In a similar way as for the DOLP difference, the spreads of distributions for cloudy classes saturate at about $L_{AT} = 0.1$, whereas it does not for the coastline class. The spread for the clear-sky pixels over land does not show a maximum as seen in the distribution of the DOLP differences.

The target polarimetric accuracy of the 3MI sensor is $5 \times 10^{-4}$ in terms of polarized reflectance over homogeneous clear sky over ocean (Fougnie et al., 2018). Figure 8 shows the fraction of pixels that satisfy this condition as a function of the along-track Laplacian (black points and lines). The blue points and lines indicate the fraction of pixels that satisfy the POLDER specification, which is $1 \times 10^{-3}$ in terms of polarized normalized radiance. The first point represents the data with $L_{AT} < 0.005$, and the interval is 0.005 up to 0.01 (21st bin). Given that 68.2 % of data fall within the $\pm 1$ standard deviation when the error distribution follows the normal distribution, we could interpret that the specification is well satisfied in the bin with the fraction larger than 68.2 % (green horizontal line). Over very homogeneous scenes with low along-track Laplacian, indeed we find that the requirements are satisfied for both 3MI and POLDER, but obviously not over as the along-track Laplacian (i.e., inhomogeneity of the scene) increases. This result is consistent with the study by Fougnie et al. (2007), which shows that the POLDER data over homogeneous scene satisfy the requirement.

## 4 Discussions

### 4.1 Prediction of error by the Monte Carlo model

The error estimation for the 3MI sensor is possible because the SGLI sensor has a similar swath and orbit as the 3MI with higher spatial resolution and less anticipated motion-induced errors. However, not all sensors have such corresponding sensor to be used for the error estimation. This is our motivation to develop a Monte Carlo model to predict the motion-induced error over cloud-over-water class, which has the highest $L_p$ error and the normalized radiance structure can be randomly generated using the power-law spectrum and inverse Fourier transform. Technical detail of the method is described in Appendix A.

Figure 9 shows the predicted magnitude of error based on the Monte Carlo simulation for cloudy pixels over water. The simulation predicts the median (red curves) that matches well to the estimation based on the SGLI data (dotted curves). As

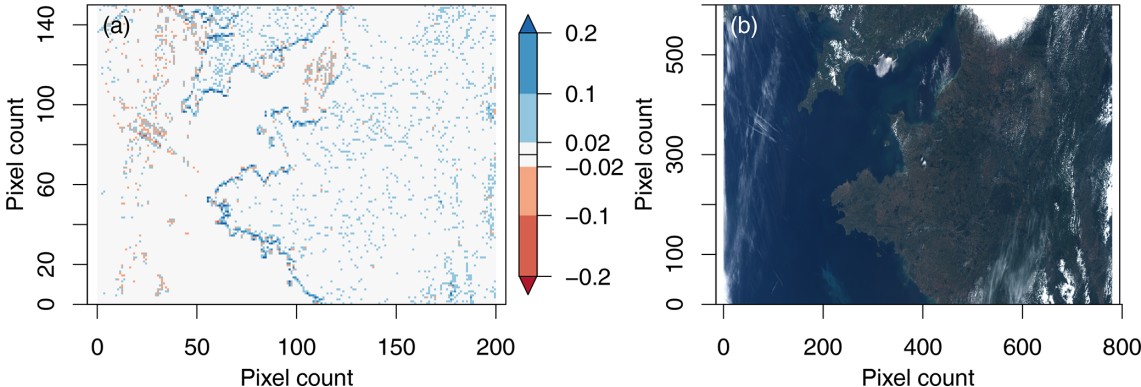

**Figure 3. (a)** The DOLP difference between proxy and reference data. **(b)** The visible composite of the SGLI Level 1B data in the same zone (visualized by authors, original data by JAXA).

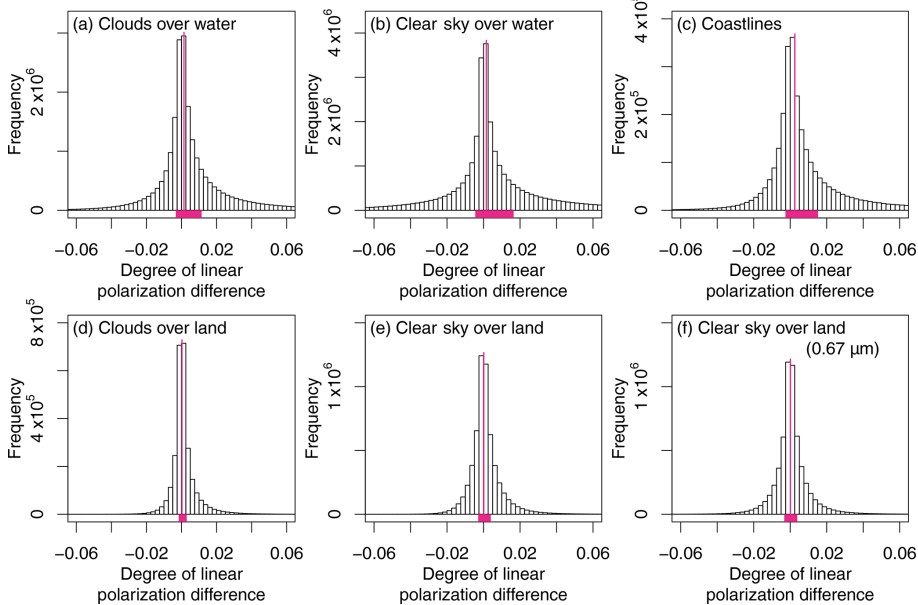

**Figure 4.** Histograms of proxy–reference differences in degree of linear polarization (DOLP) for **(a)** clouds over water, **(b)** clear sky over water, **(c)** coastlines, **(d)** clouds over land, **(e)** clear sky over land, and **(f)** clear sky over land at 0.674 μm. Except for **(f)**, the wavelength is at 0.869 μm. The vertical magenta line indicates the location of the median, and the magenta bar at the bottom indicates the interquartile range.

the empirical 95th percentile of the $L_{AT}/L$ is 0.39 and that of $L_{AT}$ is 0.070, the abscissas of Fig. 9 cover the large portion of the plausible range of the data. A slight overestimation of the DOLP difference occurs in the entire range, but the maximum difference is about $1 \times 10^{-3}$ (i.e., 2 % of the median DOLP, 0.041) in the bin $0.32 < L_{AT}/L < 0.36$. The difference between the predicted and estimated normalized polarized radiance is less than $10^{-3}$ in $L_{AT} < 0.1$. The difference reaches $6.1 \times 10^{-4}$ (i.e., 4.3 % of the median $L_p$, 0.0032) in the bin $0.09 < L_{AT} < 0.1$. Overall, the simulation can predict the median of the error estimation at better than 5 % accuracy.

The successful prediction of the motion-induced error by the statistical model implies that the error distribution in the cloud-over-water class inferred from the SGLI data is a result of the inherent horizontal structure of clouds. The method is likely to be applicable to other spatial scales to predict the magnitude of motion-induced error of the past, current, and future polarimetric instruments sharing similar designs. In the application to other satellite data, it is necessary to collect four statistics that are used in the simulation, namely, (1) normalized radiance distribution, (2) normalized-radiance-binned subpixel inhomogeneity (weighted variance), (3) normalized-radiance-binned DOLP, and (4) normalized-radiance-binned AOLP (angle of linear polarization). However, most of these statistics are readily available from observations (past and current sensors) or simulations (future sensors). The exception is the subpixel inhomogeneity, which may be a challenging parameter

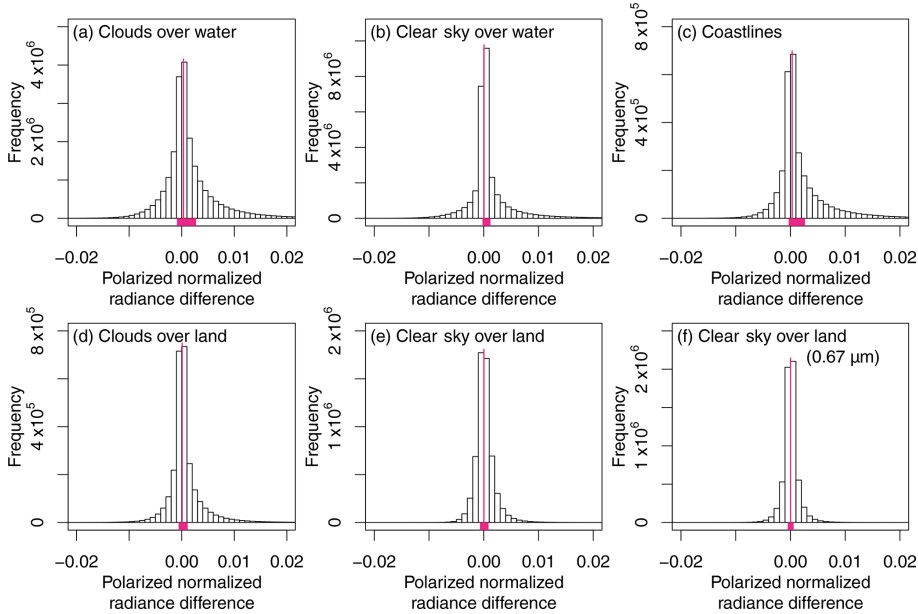

**Figure 5.** Histograms of proxy–reference differences in polarized normalized radiance ($L_p$) for **(a)** clouds over water, **(b)** clear sky over water, **(c)** coastlines, **(d)** clouds over land, **(e)** clear sky over land, and **(f)** clear sky over land at 0.674 μm. Except for **(f)**, the wavelength is at 0.869 μm. The vertical magenta line indicates the location of the median, and the magenta bar at the bottom indicates the interquartile range.

**Table 5.** Tests for the cloud detection algorithm over land. The symbols are the same as in Table 4.

| Test condition | Test type | Weight |
|---|---|---|
| ($I_{1630}/I_{868.5} < 1.2$) and ($BT_{10.8} < 288\,K$) and ($I_{673.5} > 0.2$) | Cloudy | 1 |
| ($I_{868.5} > 0.1$) or ($\sigma_{868.5} > 0.005$) | Cloudy | 10 |
| $I_{1630} > 0.1$ | Cloudy | 100 |
| ($BT_{10.8} - BT_{12} < -1\,K$) and ($BT_{10.8} < 300\,K$) | Cloudy | 1000 |
| $I_{1380} > 0.01$ | Cloudy | 1000 |
| ($I_{1630}/I_{868.5} > 1.3$) and ($BT_{10.8} > 300\,K$) | Clear sky | 1 |
| ($I_{673.5}/I_{868.5} < 0.7$) and ($I_{868.5} < 0.05$) | Clear sky | 10 |
| $0 < \sigma_{868.5}/I_{868.5} < 0.01$ and $I_{868.5} < 0.1$ | Clear sky | 100 |

to obtain if a past or current sensor has no colocated high-resolution imaging measurements, but one could replace it with an alternative measure of the local inhomogeneity, e.g., the variance of $X_0$ in the along-track direction. The modeling technique in this work is therefore particularly helpful for the preparation for future missions.

## 4.2 Sensitivity of the Monte Carlo simulation to the assumed power-law parameter

In this study, our simulation of the error statistics assumes power-law coefficient (−5/3), but it is known that the power-law coefficient varies depending on the type of clouds and spatial scale. A well-known example is for the scale break that occurs in the Landsat radiance data over stratocumulus clouds. When the spatial scale is less than a few hundred meters, the power-law coefficient becomes significantly smaller (i.e., the spectrum becomes steeper). Smaller power-

law coefficient means that low-frequency structures pronounce more than high-frequency structures. As a result, when a lower power-law coefficient is specified, the simulated 2-D normalized radiance field looks more horizontally homogeneous, introducing a smaller error in simulated 3MI polarimetric measurements.

To test the sensitivity of our prediction to the specified power-law coefficient, we simulated the error statistics for two other extreme cases. Figure 10 shows the magnitude of error in the DOLP and $L_p$ when no correlation between neighboring subpixels is assumed, and Fig. 11 shows the magnitude when a smaller power-law coefficient (−3.0) is assumed.

When there is no spatial correlation in the cloud intensity, the magnitude of simulated error is larger than the estimates based on the SGLI data. The median of the SGLI-based estimate is out of the interquartile range of the simulation (gray shading in Fig. 10) for a significant range of $L_{AT}/L$ and $L_{AT}$.

**Table 6.** The intervals of degree of linear polarization (DOLP) difference and polarized normalized radiance ($L_p$) difference for percentiles that cover 90 % and 50 % of the entire data.

| Percentile range | Pixel class | DOLP difference | $L_p$ difference |
|---|---|---|---|
| [5, 95] | Cloud over water | $[-0.023, 0.060]$ | $[-0.0051, 0.012]$ |
| | Clear sky over water | $[-0.043, 0.093]$ | $[-0.0040, 0.0088]$ |
| | Coastline | $[-0.019, 0.082]$ | $[-0.0033, 0.012]$ |
| | Cloud over land | $[-0.0075, 0.014]$ | $[-0.0033, 0.0062]$ |
| | Clear sky over land | $[-0.011, 0.016]$ | $[-0.0023, 0.0032]$ |
| | Clear sky over land (0.674 µm) | $[-0.014, 0.017]$ | $[-0.0021, 0.0023]$ |
| [15.9, 84.1] | Cloud over water | $[-0.0074, 0.022]$ | $[-0.0020, 0.0049]$ |
| | Clear sky over water | $[-0.012, 0.034]$ | $[-0.00095, 0.0027]$ |
| | Coastline | $[-0.0062, 0.029]$ | $[-0.0011, 0.0049]$ |
| | Cloud over land | $[-0.0030, 0.0055]$ | $[-0.0012, 0.0023]$ |
| | Clear sky over land | $[-0.0051, 0.0070]$ | $[-0.0011, 0.0015]$ |
| | Clear sky over land (0.674 µm) | $[-0.0059, 0.0070]$ | $[-0.00090, 0.00100]$ |
| [25, 75] | Cloud over water | $[-0.0033, 0.011]$ | $[-0.00090, 0.0027]$ |
| | Clear sky over water | $[-0.0045, 0.016]$ | $[-0.00023, 0.0012]$ |
| | Coastline | $[-0.0026, 0.015]$ | $[-0.00034, 0.0027]$ |
| | Cloud over land | $[-0.0016, 0.0030]$ | $[-0.00058, 0.0012]$ |
| | Clear sky over land | $[-0.0029, 0.0040]$ | $[-0.00069, 0.00087]$ |
| | Clear sky over land (0.674 µm) | $[-0.0032, 0.0039]$ | $[-0.00050, 0.00057]$ |

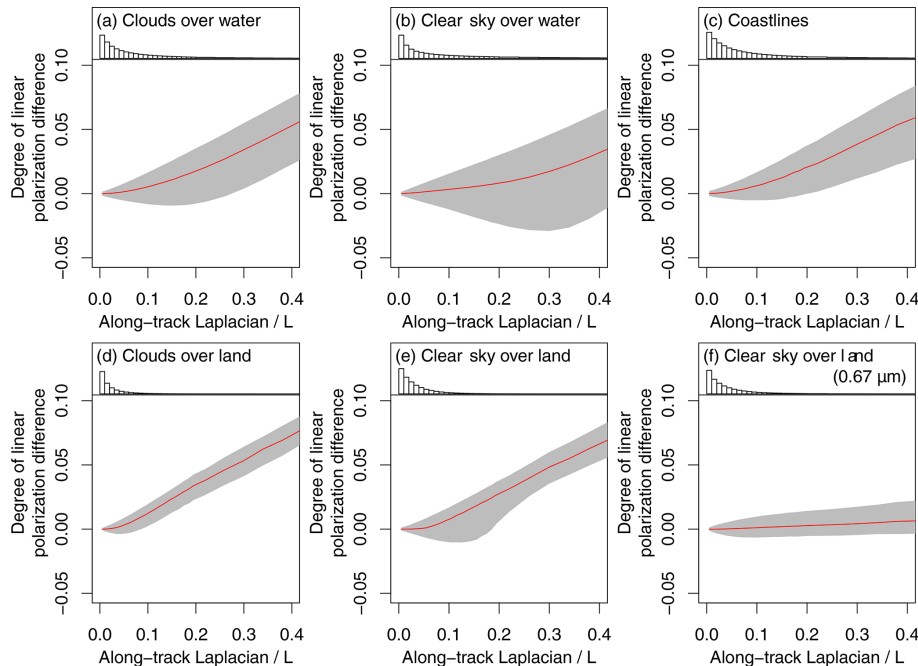

**Figure 6.** The proxy–reference differences in degree of linear polarization as a function of the along-track Laplacian divided by normalized radiance. The six panels are for **(a)** clouds over water, **(b)** clear sky over water, **(c)** coastlines, **(d)** clouds over land, **(e)** clear sky over land, and **(f)** clear sky over land at 0.674 µm. Red curves represent the median, and gray shading indicates the interquartile range. Except for **(f)**, the wavelength is at 0.869 µm.

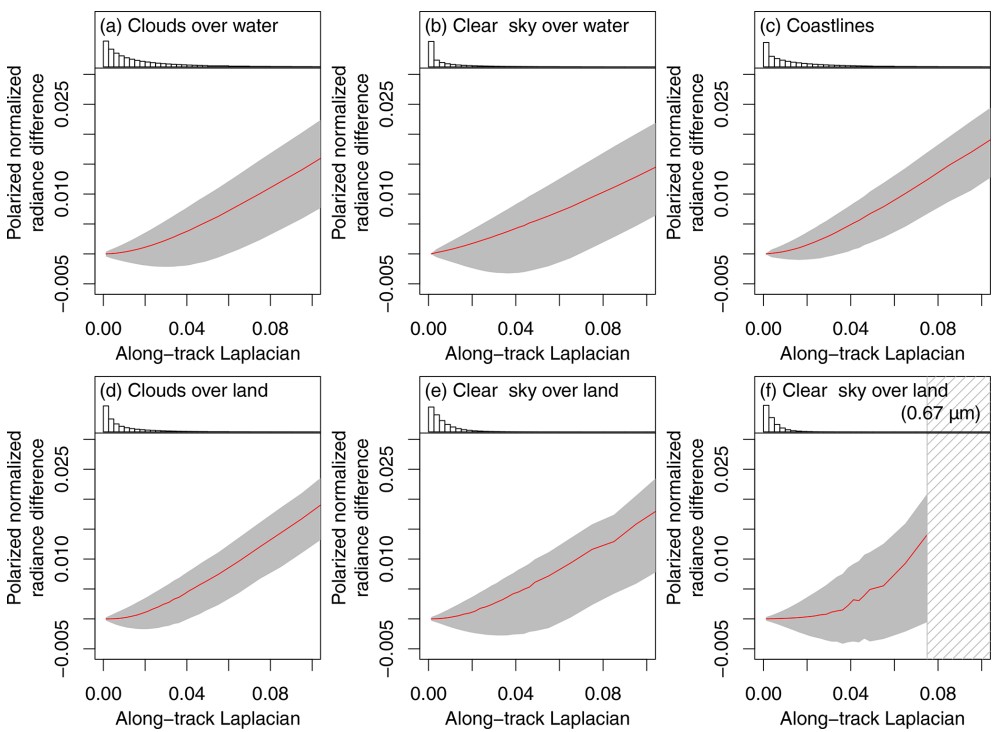

**Figure 7.** The proxy–reference differences in polarized normalized radiance as a function of the along-track Laplacian. The six panels are for **(a)** clouds over water, **(b)** clouds over land, **(c)** coastlines, **(d)** clear sky over water, **(e)** clear sky over land, and **(f)** clear sky over land at $0.674\,\mu m$. The top histogram shows the data density along the $x$ axis. Red curves represent the median, and gray shading indicates the interquartile range. Gray hatching corresponds to the part where statistics are unavailable or unreliable because of a limited number of data points. Except for **(f)**, the wavelength is at $0.869\,\mu m$.

**Table 7.** The medians of degree of linear polarization (DOLP) difference and polarized normalized radiance ($L_p$) difference.

| Pixel class | DOLP difference | $L_p$ difference |
|---|---|---|
| Cloud over water | 0.0014 | 0.00035 |
| Clear sky over water | 0.0015 | 0.000049 |
| Coastline | 0.0025 | 0.00031 |
| Cloud over land | 0.00027 | 0.000089 |
| Clear sky over land | 0.00014 | 0.000037 |
| Clear sky over land ($0.674\,\mu m$) | 0.000099 | 0.000020 |

On the other hand, Fig. 11 shows that the magnitude of simulated error is slightly lower when a smaller power-law coefficient ($-3.0$) is assumed. In addition, we note that the SGLI error estimation for the clouds-over-land category also shows the slope between these two extremes.

From these results, we propose that the spatial correlation of intensity due to the natural cloud structure should be considered when predicting the statistics of motion-induced error over clouds. For the 3MI sensor, the higher resolution observations provided by the METimage sensor (Wallner et al., 2016) aboard the same MetOp-SG-A platform might be used to further constrain the along-track scene spatial correlation

and therefore be useful to improve the pixel-level polarimetric uncertainty estimates.

### 4.3 Application of the results

In this study, we estimated 3MI's motion-induced error from the SGLI data. A straightforward implication of the correlation between the motion-induced errors and the along-track Laplacian is that it is possible to predict the quality of polarization data in every pixel. For example, from the statistics we obtained in this study, the median bias and the standard deviation for every pixel can be provided as a function of $L_{AT}$, which is a derivable quantity through $X_0$. The challenge of this approach is that the view angle of the SGLI is fixed at positive or negative $45°$ in the along-track direction and therefore representative of only a part of 3MI's view direction. However, as the error estimate shows, the spatial inhomogeneity primarily determines the motion-induced error, and statistics obtained in this study are still useful. In addition, the intensity variation over the land is primarily due to the land surface inhomogeneity, and month-long or year-long statistics would be beneficial to define the standard motion-induced error for aerosol retrieval purposes.

We also confirmed in this study that the motion-induced error over the cloud-over-water class can be predicted with

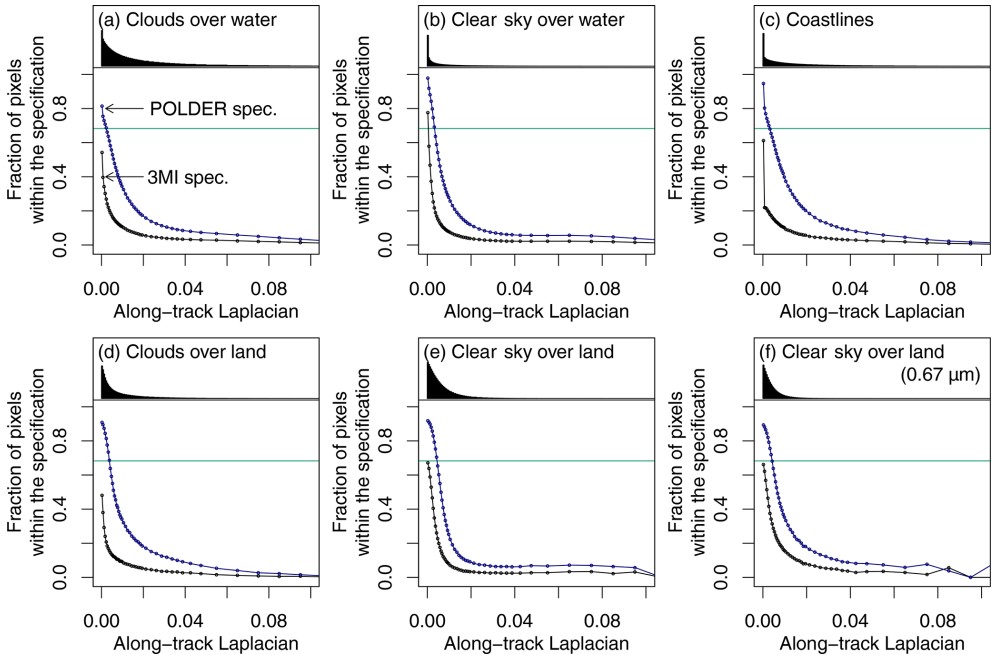

**Figure 8.** The fraction of pixels within the POLDER specification (blue) and the 3MI specification (black) in each bin of along-track Laplacian. The six panels are for **(a)** clouds over water, **(b)** clear sky over water, **(c)** coastlines, **(d)** clouds over land, **(e)** clear sky over land, and **(f)** clear sky over land at 0.674 μm. The density histograms of the along-track Laplacian are presented on the top of each panel. The green horizonal line shows the fraction of 0.682, which corresponds to the fraction of data within $1\sigma$ of a normal distribution.

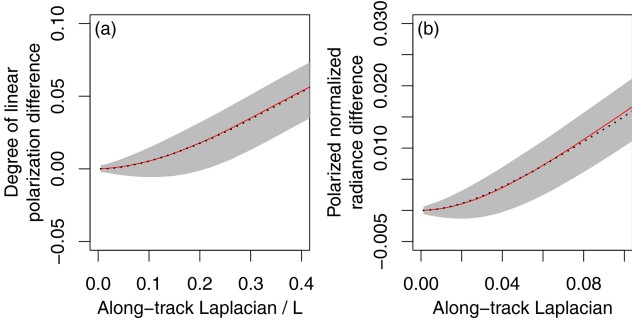

**Figure 9. (a)** The simulated proxy–reference difference in the degree of linear polarization as a function of the along-track Laplacian divided by normalized radiance **(b)** The simulated proxy–reference difference in the polarized normalized radiance as a function of the along-track Laplacian. Red curves represent median, and gray shading indicates the interquartile range of the simulation. Black dotted curves correspond to the medians of the observation (i.e., red curves in Figs. 6a and 7a).

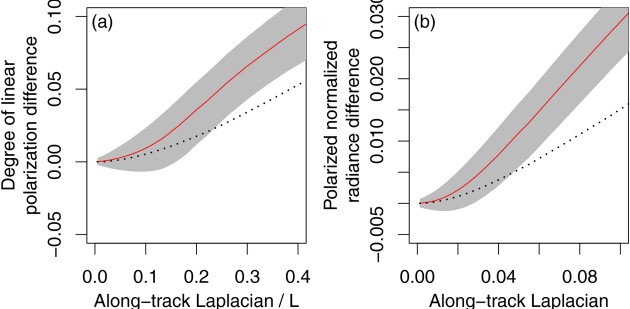

**Figure 10.** The same as Fig. 8 but without correlation in the cloud field simulation.

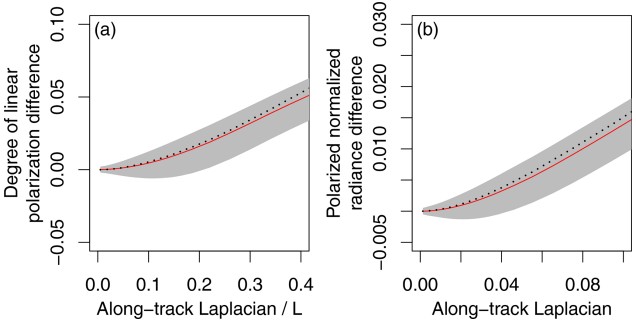

**Figure 11.** The same as Fig. 8 but with slope of −3.0.

a Monte Carlo model. The advantage of the simulation is that it does not require high-resolution polarimetry to perform the error analysis. In the application to the 3MI data, among the four required input statistics, three statistics other than cloud subpixel inhomogeneity are readily available from the 3MI sensor itself. The cloud inhomogeneity parameter can be provided either from the colocated METimage sensor

or by replacing with alternative measure of the inhomogeneity, exploiting the stable reflectivity power spectrum in the few kilometers scale. As the motion-induced $L_\mathrm{p}$ error for the cloud-over-water class was the largest, the simulation technique is helpful to determine the upper limit of the motion-induced $L_\mathrm{p}$ error.

## 5   Conclusions

From the high-resolution global polarimetric observation by the SGLI, we estimate 3MI's polarimetric error induced by the co-registration and interpolation that compensates for the satellite's along-track motion during the acquisition. The estimates show that the magnitude of the interpolation error is not negligible nor symmetric about the zero, particularly over the locations where intensity variation is large.

The asymmetric distribution of the motion-induced error is confirmed in all four categories of analyzed pixels: clear sky over land, clouds over land, clouds over water, and coastlines. The clear-sky-over-water class had the most spread distribution of the degree of linear polarization (DOLP) difference, whereas the cloud-over-water class had the most spread distribution of the polarized normalized difference ($L_\mathrm{p}$) differences. The 9th to 95th percentile range was $[-0.019, 0.082]$ for the DOLP differences and $[-0.0051, 0.012]$ TS1 for the $L_\mathrm{p}$ differences. These differences strongly depend on the along-track Laplacian that characterizes the nonlinear change in the normalized radiance ($L_\mathrm{p}$ difference) or that divided by the normalized radiance (DOLP difference). As the along-track Laplacian is available from the observation, a pixel-level quality information can be obtained based on the statistics presented in this study. In addition, it is possible to perform the statistical estimation of error when the subpixel inhomogeneity is available as an input. Either procedures proposed in this study would help ensure the reliability of the radiance products and downstream applications including retrieval products as well as data assimilation.

The error statistics based on the SGLI data and Monte Carlo simulation agree within 5 % in terms of median bias, implying the predictability of the error for cloudy pixels at an arbitrary spatial scale. The sensitivity study on the assumed power-law coefficient proved the need for the spatial correlation to be included in the error prediction. The method of prediction is applicable to past, current, and future missions with a polarimetric instrument based on a similar design, paving a way to better predict the performance of the instrument on orbit at the stage of planning. Finally, although simulations presented here are limited to cloudy scenes, a similar diagnosis might be made for land surfaces under clear-sky conditions for use in aerosol remote sensing applications.

## Appendix A: Monte Carlo model of the motion-induced error

The statistics of the error are simulated with a Monte Carlo model for pixels in the cloud-over-water class. In summary, we first generate synthetic 2-D normalized radiance fields by the inverse 2-D Fourier transform, assuming the power-low spectrum of the normalized radiance distribution. We then proceed with the weighting and aggregation equivalent to the method described in Sect. 2.1. This process is repeated for 10 million samples to realize meaningful statistics. The details of the method are described in the rest of this Appendix.

The first step is to apply an inverse 2-D Fourier transform to the assumed power-law spectrum with Gaussian noise as in the method described by Iwabuchi and Hayasaka (2002). A number of studies show that the intensity spectrum of the cloud field follows the power law (Cahalan and Snider, 1989; Davis et al., 1997; Marshak et al., 1995, 1998; Oreopoulos et al., 2000), and the slope is known to be a function of horizontal spatial scale. For scales greater than a few hundred meters and less than a few hundred kilometers, the power-law coefficient is between $-1.5$ and $-2.0$, and the value increases for the smaller. We select $-5/3$ ($1.666\ldots$) as the power-law coefficient in our simulation, as it is the value commonly referred to. The impact of this choice is evaluated by the simulations with varied parameter in Sect. 4.2.

Every synthetic 2-D normalized radiance field that is generated in this way is different. They are, however, statistically centered at zero, and their variance depends on the prescribed power-law coefficient. In the second step, to make the simulation consistent with the SGLI-based estimation, we scale and add an offset to each realization as in the following equation:

$$L_{(i,j)} = ax_{(i,j)} + b, \tag{A1}$$

where $L_{(i,j)}$ is the final 2-D normalized radiance field, $x_{(i,j)}$ the output of the inverse Fourier transform, $a$ the scaling coefficient, and $b$ the offset. The scale $a$ and offset $b$ are determined so that the pixel intensity $\overline{L} = \sum_i w_i L_i$ and weighted variance $V = \sum_i w_i (L_i - \overline{L})^2$ follow the observed empirical distribution function (EDF) of normalized radiance $\hat{F}_L(l)$ and weighted variance $\hat{F}_V(v; l)$. These EDFs are computed from the SGLI data upon performing the error estimation described in Sect. 2.1. The EDF of variance $\hat{F}_V(v; l)$ is computed for each small normalized radiance interval, spanning from 0 to 0.9 with 0.01 width, from 0.9 to 1 with 0.05 width, and from 1 to 1.5 with 0.5 width. The weight $w_i$ is the arithmetic mean of final weights in Table 3 for every line (i.e., for line 5, it is $1/16 + 9/320 + 11/320 = 1/240$).

The third step is to compute $X_{1(i,j)}$, $X_{2(i,j)}$, and $X_{3(i,j)}$ from the 2-D normalized radiance field $L_{(i,j)}$; the angle of linear polarization (AOLP) $\chi$; and the degree of linear polarization (DOLP) $\delta$. The AOLP is sampled so that it follows the EDF of AOLP $\hat{F}_\alpha(\alpha; l)$, and the DOLP is sampled so

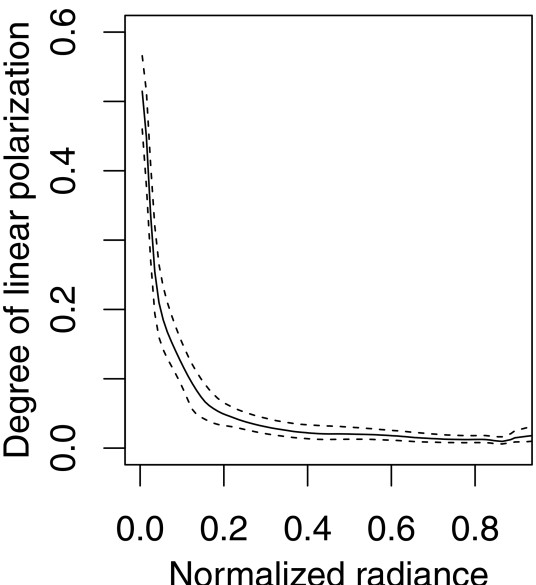

**Figure A1.** Decreasing trend of the degree of linear polarization for pixels over water with increasing intensity. The solid line is median, and dashed lines are 25th and 75th percentiles (interquartile range).

that it follows EDF of DOLP $\hat{F}_\delta(\delta; l)$. The AOLP is assumed constant over subpixels for each realization, but the DOLP is sampled for each subpixel because of the strong intensity dependence with weak spatial correlation. Figure A1 shows the strong decreasing trend of the DOLP with intensity. We note here that the statistics of the intensity-binned DOLP and the statistics of AOLP have been found necessary for the reasonable simulation of the polarimetric error. This is presumably because the DOLP varies significantly as a function of reflectivity at cloud boundaries and over thin clouds, and the AOLP determines the relative contribution of error in $X_1$ and $X_3$ images to the polarized normalized radiance ($L_p$).

Once $X_{1(i,j)}$, $X_{2(i,j)}$, and $X_{3(i,j)}$ are obtained, the last step is to proceed through with the weighting and aggregation equivalent to the method described in Sect. 2.1. This entire process of 2-D Fourier transform, scaling, and $X_1 - X_2 - X_3$ derivation is repeated 10 million times.

*Data availability.* GCOM-C/SGLI data used in this study are available from the G-Portal website and ftp service operated by Japan Aerospace Exploration Agency (ftp://ftp.gportal.jaxa.jp/standard/GCOM-C/GCOM-C.SGLI/L1B/1/ TS2) (Japan Aerospace Exploration Agency GCOM-C project team, 2018b) TS3. The user registration is required through the website (https://gportal.jaxa.jp/gpr/ TS4) to access the data.

*Author contributions.* SH and JR conceived of the methodology and outlined the project. SH performed the study and composed the article. JR encouraged and supervised the project. MSD contributed the research activities through discussions on the results.

*Competing interests.* The authors declare that they have no conflict of interest.

*Acknowledgements.* We appreciate Japan Aerospace Exploration Agency (JAXA) for providing the GCOM-C/SGLI data needed for the research. This research is conducted in a framework of jointly funded postdoctoral fellowship by Centre National d'Études Spatiales (CNES) and the Make Our Planet Great Again (MOPGA) program by Campus France. The authors are grateful for the financial support and various encouragements by CNES, MOPGA, and the Laboratoire d'Optique Atmosphérique. Our gratitude extends toward Bertrand Fougnie, who provided us valuable comments regarding the 3MI and POLDER mission requirements. It is our pleasure to hereby express our gratitude to three reviewers, Frans Snik, Otto Hasekamp, and Kirk Knobelspiesse, for their constructive and positive comments.

*Financial support.* This work was supported by CNES through the TOSCA program under grant no. 5413/MTO/4500065502.

*Review statement.* This paper was edited by Andrew Sayer and reviewed by Otto Hasekamp, Kirk Knobelspiesse, and Frans Snik.

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

**Remarks from the typesetter**

TS1    Please give an explanation of why this needs to be changed. We have to ask the handling editor for approval. Thanks.

TS2    Please check the URL.

TS3    Please confirm.

TS4    Please provide date of last access.

TS5    Please note that this reference is not mentioned in the text.