# Peer review of "A study of polarimetric error induced by satellite motion: Application to the 3MI and similar sensors"

_Atmospheric Measurement Techniques, 2020_

## Referee Comment (RC1) · Frans Snik (Referee) · 2 Nov 2020

The authors present a simulation of future 3MI polarimetric data based on existing and modeled data, and assess the impact of sequential polarization-filtered intensity measurements along the ground track. This is indeed a major limiting factor to the polarimetric uncertainty of this and similar instruments, and therefore deserves an in-depth analysis and publication in AMT.

However, this reviewer wants to raise a number of fundamental and more minor aspects that should first be addressed in more detail:

[Figure]

1) The uncertainties discussed in the manuscript are purely systematic and not at all random, and should therefore not be labeled as "noise". Moreover, it is debatable to what extent a statistical framework can accurately describe the uncertainties due to these systematics, as there are no random processes involved in the measurement itself. In addition, as DoLP is necessary larger than (or equal to) zero, any distributions for low DoLP will be skewed. It is only mentioned in passing, but to a large extent these systematic effects can be predicted and therefore partially mitigated, as the spatial structure that induces them is always measured. This should be discussed further.

2) It remains unclear why the Laplacian is used as a proxy for the systematics instead of a regular gradient. Indeed, the gradient literally constitutes the first-order approximation. But, as Stokes parameters are obtained through intensity differences, spurious polarization effects are readily described as the amount of uncorrected shift times the gradient in the flight direction + higher-order terms. Or do you assume that the shifts are fully corrected, and that the first non-linear term is the leading one? Can you demonstrate this by propagating a simple model through the polarimetric demodulation?

3) The other fundamental limitation of this polarimetric method is not discussed at all: Remaining pixel-to-pixel gain variations can also lead to spurious polarization signals, which are indeed more noise-like. What do you assume for the flat-fielding calibration accuracy? And why not take this into account in the simulation as well? Indeed, sub-pixel effects are mentioned, and may need to be quantified with a back-of-the-envelope propagation as well.

Detailed comments:

-It would be highly useful to list the polarimetric requirements for 3MI to put these results into context.

-Same for the atmospheric parameters that are derived from these measurements.

-l37: It would be good to discuss the basics of these vicarious calibration methods. Can they be applied to mitigate for the effects discussed in this paper?

-There is a mistake in Eq. 5: The last factor should be X_m60.

-Why not consider the Angle of Linear Polarization?

-And why not stick to the Stokes parameters [Q,U] or [Q/I, U/I] to keep things mathematically well-behaved?

-I don't understand the sentence on lines 69-70... Maybe you should distinguish additive and multiplicative spurious polarization effects?

-Please discuss in detail the commonalities and differences hinted at in Sect. 2.1 between the polarimetric implementations of SGLI and 3MI. A cartoon figure may come in handy.

-Please explain the particular values in Tables 1 and 2.

-l100: Please provide some more details on what you interpolate, and how.

-Why is there a difference in "center of mass" between the shifted grids in Tab. 2? Does this have an impact on the spurious polarization signals?

-Sect. 2.2.2 and further: I don't understand the meaning of "stratification" in this context...

-As the cloud detection algorithm is new, it would be good to provide some numbers on false positives/negatives.

-l113: Introduce all the acronyms.

-What are the confidence levels for cloud (non-)detections?

-I suppose there is a valid reason why you select a power law of -5/3 (Kolmogorov)?

-l145: Please provide a reference for the "observed empirical distribution function".

-l150-158. Why not compute this in [Q,U] over which you can actually average? If the AoLP is not constant, this math breaks down. . .

-l202: I don't understand "0.0010 (i.e. 2.2%)".

-It would be insightful to also present intensity and polarimetric images for typical simulated cloud scenes for particular power law distributions.

---

## Referee Comment (RC2) · Otto Hasekamp (Referee) · 3 Nov 2020

This paper investigates the polarimetric error that results from non-instantaneous acquisition by different polarization filters, representative for filter-wheel based instruments such as 3MI and the POLDER-1,-2, and -3 instruments. It has been suggested for long time in the polarimetric remote sensing community that this may be an important source of error and I'm glad to see that the authors quantify this error term. Also, the authors show that the magnitude of the polarimetric error may be predicted using the along-track Laplacian, available from the measurements themselves. This provides important input for retrieval algorithms of aerosol and cloud properties. I have

a number of comments which are important but quite easy to address. I recommend publication of the paper after addressing the comments.

Comments

- It would be better to use the term 'error' instead of noise because it is not a purely random error (see also comment of other reviewer). Also, I suggest to clearly quote values for bias and RMSE as measures of accuracy. The 5-95 percentile is also useful but the inter-quartile range is less relevant. Probably here the 15.9-84.1 percentile range is more useful as it would represent the standard deviation for a Gaussian error distribution.

- Please quote the bias and RMSE for all 4 categories in the abstract.

- Which wavelength has been used for this study? Was it 670 or 860 nm? For clear-sky over land I would expect the error at 860 nm is (much) larger than at 670 nm. I would recommend to show errors at both wavelengths.

- Please make clear that the resulting errors are on L1B product. Further processing to L1C may introduce extra errors (Lang et al., JQSRT, 2019), but not necessarily related to the measurement acquisition approach.

- Please relate the errors to the total error budget of 3MI. Is it a dominant error source?

- How do these errors relate to errors on aerosol and cloud properties. For aerosols, we have investigated the relation between DoLP accuracy and retrieval accuracy (e.g. Hasekamp et al., JQSRT, 2019). Not sure if a similar study exists for cloud retrieval.

- Also the case of clear-sky over ocean would be interesting. Probably the errors would be small here (at 670 and 860 nm, maybe not at 400-500 nm) but that would also be important information for users. I suggest to add that case.

- Figure 6 shows the median and 'spread' of the error as function of L_at/L. Can the median be interpreted as the median of the DoLP_diff distribution corresponding to a

given value of L_at/L? I would expect this is a bin of L_at/L values, otherwise you would have very few points to compute the mean and spread. Or am I mis-interpreting the figure? How is spread defined here? - Why are there large parts in Fig 7 'Gray hatched' (statistics unavailable or unreliable) but not in Fig. 6?

- The authors show that the noise found from SGLI for the cloud over water case is similar to the noise found for a radiance field simulated for random cloud distributions following a power law distribution. They call this 'prediction of noise'. I would say this is not the 'prediction of noise/error' , but rather an 'understanding of error'. The prediction of error comes from the correlation with L_at/L in Fig. 6 with real data. The results from the Monte Carlo simulations are interesting to understand the results but hamper the flow of the paper. I suggest to move this part to an Appendix.

———————————————

---

## Referee Comment (RC3) · Kirk Knobelspiesse (Referee) · 5 Nov 2020

Review of "A study of polarimetric noise induced by satellite motion: Application to the 3MI and similar sensors" by Hioki et al., amt-2020-407

This manuscript characterizes the error of observations by rotating element polarimetric sensors such as 3MI and POLDER. While this has long been known as a potential issue with this type of instrument, this manuscript provides a reliable quantification of these errors, and a description of potential systematic biases as well.

In terms of scientific significance and quality, this manuscript is excellent. There are

a few (minor) issues regarding presentation quality and terminology that are easily addressed.

More detailed comments and questions.

1. I'd like to see a clearer description in the abstract and introduction that the co-registration and interpolation issue in 3MI (and POLDER) is due to the non-simultaneous observation of polarization filtered intensity measurements (Xm60, X0, Xp60) inherent to rotating filter wheel style instruments. This is important because some other polarimeters have made the (sometimes expensive) choice to use division of aperture or focal plane or other techniques to avoid this issue, and as such have lower expected uncertainties (which generally have less systematic bias too). It is an important point to make for readers who may not be as immersed in the polarization community as ourselves. It would also help explain why SGLI is the reference without interpolation issues. I realize you allude to this issue in several parts of the paper, but I'd like to see a clearer indication in the abstract and start of the introduction.

2. I'm not sure "noise" is the right term to use throughout your paper, since it often involves a bias, so it isn't 'random'. You cite the Povey and Grainger paper, and to be consistent with that you should use the term 'error'.

3. Section 2.2.2: is there a reference for the meaning of the file format names that aren't spelled out (POLDK, VRNDK, VNRDL, IRSDK, etc.)

4. Section 2.2.2 – I would think 'classification' of the data, not stratification, is a more appropriate term. Later on you use the term stratification in a different way too, and I don't think it is correct their either (more on that in a bit).

5. Section 3.1, Fig 3a I'm not sure I follow why the coastlines have what appears to be a significantly biased error (blue in Fig 3a) but the clouds do not. I'm also confused why we don't see these errors for clouds over land in the figure. Perhaps that is a function of the color scaling, which needs to be tightened significantly to make the errors more

obvious. This figure is very hard to see.

6. As an aside, how would cloud motion during the filter wheel acquisition period, affect the results? I realize that is not incorporated into this study, but the techniques of this study could be used to address that issue too, perhaps.

7. Results and Figure 6-10. Here the word stratified is used as well, when I think you really mean correlation, as in, the error is correlated with the Laplacian

8. I missed the explanation of way you are dividing the Laplacian by L.

9. I'm not sure the Discussion section is really any different than a part of the results.

10. What would be a nice thing to include in a discussion is some thoughts about how these results can be used. Are you suggesting it might make sense to assess the observations of the VII sensor to correct the 3MI data for expected systematic bias? Or would it make more sense simply to use this uncertainty estimate to help weight the observations in a retrieval algorithm?

11. Are the shifting weights described in Table 1 and 2 the same for all view angles? I think the answer is yes, but if not there would be implications on differential weighting of one view angle versus another.

12. It would be nice to see more discussion of how large these errors are compared to the overall 3MI uncertainty. Are they the main source of uncertainty? What is the significance of these uncertainties for the ability to retrieve geophysical parameters (you could point to information content studies here).

Overall though, great paper. Thank you

---

## Author Comment (AC1) · 23 Dec 2020

We are very grateful for the reviewer's careful and constructive comments. The comments contain insightful questions and the new perspectives that authors did not initially realize. This can be possible only through the reviewer's profound experience and knowledge, as well as attentive reading of the manuscript. Thank you very much for the time spent in providing us useful comments. The followings are the response from the authors to those comments.

**Major comment (1)-1**

[Figure]

- The uncertainties should not be labeled as "noise".

We agree with the reviewer to be precise in the terminology of the uncertainty. With suggestions from other reviewers, we replace the term "noise" with "error" and other appropriate terminology in the revised manuscript.

**Major comment (1)-2**

- It is debatable to what extent a statistical framework can accurately describe the uncertainties.

The reviewer points out that there is no random process in the mechanism of uncertainties, and therefore the statistical approach may not be the best description of the error. The authors share the same view as the reviewer, and for this reason we did not attempt to perform a parametric analysis of the distribution. The distributions (histograms and correlations) are presented to better describe the nature of the error, rather than to fit the distribution with a known simple distribution.

**Major comment (1)-3**

- Any distributions for low DOLP will be skewed.

For sure this is true, and the authors would like to add that any distributions for high DOLP will also be skewed.

**Major comment (1)-4**

- To a large extent, these systematic effects can be predicted and therefore partially mitigated, as the spatial structure that induces them is always measured.

There are two possible measurements of the spatial structure in the case of the 3MI. First is the measurement from the 3MI itself, and the other is the one from the METimage instrument that is also on the same satellite platform. The 3MI has 4 km resolution, whereas METimage has higher resolution of 500 m. The first approach, which uses only the 3MI data, is the main interest of this study. As shown in the response to the Major comment (2) below, our understanding is that the effect of satellite motion is already mitigated to the first order by the linear interpolation, and our discussion is primarily on the residual error. Non-linear interpolation might indeed improve the mitigation, but it is at the expense of the spread in the modulation transfer function (i.e. blurred image). For this part, the authors admit that a dedicated study would be necessary based on the results that we obtained in this study. The second approach is important for the mitigation of the error, and we clarified the possible use of the METimage data in the discussion section of the revised manuscript.

**Major comment (2)**

- It remains unclear why the Laplacian is used as a proxy for the systematics instead of a regular gradient.

The authors are grateful for the reviewer's careful comment, and we are happy to take this opportunity to present what we know about the distinction between the linear term and the non-linear terms in the systematics. As the reviewer pointed out, the spurious polarization is described as the amount of uncorrected shift times the gradient + higher order terms. In other words, the values of shifted $X_{m60}$ and $X_{p60}$ can be written as follows:

$$X_{m60}(-s) = X_{m60}(0) - \frac{dX_{m60}}{dx}s + O(s^2) \tag{1}$$

$$X_{p60}(s) = X_{p60}(0) + \frac{dX_{p60}}{dx}s + O(s^2) \tag{2}$$

where s is the amount of shift in pixels ($0 \leq s < 1$). Without "unshifting", the $L_p$ is computed from $X_{m60}(-s)$, $X_0(0)$ and $X_{p60}(s)$ in place of $X_{m60}(0)$, $X_0(0)$, and $X_{p60}(0)$, and therefore contains an error. The mitigation could be performed by first computing the gradient from the two measurements near 0. Note that for $X_{m60}$, the measurements are performed at $x = \cdots, -1-s, -s, 1-s, \cdots$, and for $X_{p60}$, at $x = \cdots, -1+s, s, 1+s, \cdots$.

$$\frac{dX_{m60}}{dx} = X_{m60}(1-s) - X_{m60}(-s) + \epsilon_m \tag{3}$$

$$\frac{dX_{p60}}{dx} = X_{p60}(s) - X_{p60}(s-1) + \epsilon_p \tag{4}$$

where $\epsilon_m$ and $\epsilon_p$ are errors due to the discretization. Then, Eqs. (3) and (4) are substituted into Eqs. (1) and (2):

$$X_{m60}(-s) = X_{m60}(0) - sX_{m60}(1-s) + sX_{m60}(-s) + O(s^2) + s\epsilon_m \tag{5}$$
$$X_{p60}(s) = X_{p60}(0) + sX_{p60}(s) - sX_{p60}(s-1) + O(s^2) + s\epsilon_p \tag{6}$$

The best estimates of the corrected $X_{m60}(0)$ and $X_{p60}(0)$ could be therefore defined as follows:

$$\hat{X}_{m60}(0) = (1-s)X_{m60}(-s) + sX_{m60}(1-s) \tag{7}$$
$$\hat{X}_{p60}(0) = (1-s)X_{m60}(s) + sX_{m60}(s-1) \tag{8}$$

Equations (7) and (8) are exactly the linear interpolations that we perform to obtain the unshifted $X_{m60}$ and $X_{p60}$ in this study. The leading error terms in Eqs. (5) and (6) are $O(s^2)$ and we therefore consider that Laplacian is the most appropriate measure of the residual error. We include a brief conclusion from this calculation in the revised manuscript, as interested readers could also refer to this Authors' Response online for full details. Of course, at your recommendation, we are ready to include this discussion as an appendix.

**Major comment (3)**

- Remaining pixel-to-pixel gain variations can also lead to spurious polarization signals. What do you assume for the flat-fielding calibration accuracy? And, why not take this into account in the simulation as well?

We agree with the reviewer that the remaining pixel-to-pixel gain variation can contribute to the error. In this study, however, it is not taken into consideration. The flat-fielding calibration target for the 3MI is 2% between any two pixels in a field of

view, and 0.1% for any 10×10 pixels area (Fougnie et al. 2018). As the shift is 0.45 pixels, which is way less than 10 pixels, the 0.1% is the relevant accuracy requirement. In addition, the contribution factor of the same detector element is 55% $(1 - (1.8\text{km})/(4\text{km}) = 0.55)$ in the linear interpolation that unshift the $X_{p60}$ and $X_{m60}$ images. According to our simple estimation without spatial correlation, the 0.1% calibration error results in $_p = 8.8 \times 10^{-5}$ (median) for an originally unpolarized scene with $L = 0.2$. This is significantly smaller than the error we obtained, and the actual characteristics of the residual gain between adjacent pixel is not available. We therefore didn't include the flat-field calibration error into this study.

**Other comments**

- It would be highly useful to list the polarimetric requirements for 3MI to put these results into context.

We appreciate this suggestion very much, along with similar suggestion by other two reviewers. It is indeed important to put the results into the context of the system requirements. To put our results into wider perspectives, we're providing here a comparison to both the previous POLDER requirements and the targeted 3MI ones. The target polarimetric accuracy of the 3MI sensor is $5 \times 10^{-4}$ in terms of polarized reflectance over homogeneous clear-sky over ocean (Fougnie et al. 2018). Figure 1 shows the fraction of pixels that satisfy this condition as a function of the along-track Laplacian (black points and lines). The blue points and lines indicate the fraction of pixels that satisfy the POLDER specification, which is $1 \times 10^{-3}$ in terms of polarized normalized radiance. Given that 68.2% of data falls within the $\pm 1\sigma$ when the error distribution follows the normal distribution, we could interpret that the specification is well satisfied in a particular bin when the fraction exceeds 68.2%. Over very homogeneous scenes with low along-track Laplacian, indeed we find that the requirements are satisfied, but obviously not anymore as the along-track Laplacian (i.e. inhomogeneity of the scene) increases. This result is consistent with the study by Fougnie et al. (2007), which shows that the POLDER data over homogeneous scene satisfy the requirement. In the

revised manuscript, we add one paragraph that discuss this point.

- Same for the atmospheric parameters that are derived from these measurements.

It is also our interest how the polarimetric error from this study impact the derived atmospheric parameters. However, the extraction of the parameters depends significantly on the algorithm and assumption in the retrieval algorithms. In this study, we therefore prefer to stay focused on the error for the Level 1B products.

- It would be good to discuss the basics of these vicarious calibration methods.

We appreciate this comment. Some vicarious calibration methods are intended for the absolute calibration whereas others are more suited to transfer the calibration coefficients between wavelengths. We add in the revised manuscript a paragraph containing the characteristics of these methods.

- There is a mistake in Eq.5: The last factor should be $X_{m60}$.

Thank you very much for catching this error. We correct the equation in the revised manuscript.

- Why not consider the angle of linear polarization? And why not stick to the Stokes parameters (Q,U) or (Q/I,U/I) to keep things mathematically well-behaved.

The angle of linear polarization (AOLP) and Stokes parameters are also of an interest for certain readers. The authors agree with the reviewer that the analysis based on the Stokes parameters (Q and U) would be beneficial, particularly for the mitigation of non-linear term contributions, as Q and U are linearly related to the measured intensity. The reason why we selected the $L_p$ and the DOLP for presentation is that they are two most frequently-used parameters in the downstream applications including the retrieval of physical quantities. In addition, the AOLP error statistics is likely more sensitive to the viewing geometry difference between the SGLI and 3MI. As the SGLI covers only a part of viewing geometry of the 3MI, we prefer to present the results for DOLP and $L_p$.

- I don't understand the sentence on lines 69-70, "Rather, the noise tends to suppress the polarization for strongly polarized target and tends to enhance the polarization for weakly polarized target."... Maybe you should distinguish additive and multiplicative spurious polarization effects?

In the revised manuscript, we edited the sentence to make the point clearer. The sentence was to mean that noise for a scene with DOLP≈0 tends to increase the DOLP whereas a noise for a scene with DOLP≈1 tends to reduce the DOLP because the DOLP is bounded by 0 and 1.

- Please discuss in detail the commonalities and differences as hinted at in Section 2.1 between the polarimetric implementations of SGLI and 3MI. A cartoon figure may come in handy.

We appreciate the suggestion from the reviewer. We add a paragraph clarifying the commonalities and differences between the SGLI and 3MI in the revised manuscript. A cartoon figure as shown in Fig. 2 is added to highlight that the commonality is the measurement principle (three intensity measurements with linear polarizer at different angles) and the difference is the acquisition arrangements.

- Please explain the particular values in Tables 1 and 2.

The values in Table 1 is computed from the following equation:

$$w_0(i) = \frac{1}{16} \int_{i-1}^{i} \Pi_{4,8}(x - s)dx \tag{9}$$

where i is the index of line, $\Pi_{a,b}(x)$ is a boxcar function that is 1 in the interval $(a,b)$ and 0 otherwise, and s is the amount of shift in SGLI pixel size (i.e. +1.8 or -1.8 in our case). This is an integral of a boxcar function in the field-of-view with a boxcar modulation transfer function. The values in Table 2 is due to the consequence of the linear interpolation near the center of image, and can be computed as follows:

$$w(i) = (1 - \frac{s}{4})w_0(i) + \frac{s}{4}w_0(i + 4\mathrm{sgn}(s)) \tag{10}$$

where $\mathrm{sgn}(s)$ is sign of the shift.

- l100 Please provide some more details on what you interpolate, and how.

The reviewer points out that it is uncertain what is interpolated in the unshifting. At this stage of data processing, we have the $X_{m60}$ and $X_{p60}$ by aggregating $4 \times 4$ SGLI pixels, but these images are shifted by $\pm 1.8$ SGLI pixel into the along-track direction to mimic the 3MI shift caused by sequential acquisition. For this reason, linear interpolation is necessary to obtain the $X_{m60}$ and $X_{p60}$ at the pixel centers of $X_0$. In the revised manuscript, we make this point clearer. We hope that the addition of the equation in response to the previous question also helps to understand what is performed.

- Why is there a difference in "center of mass" between the shifted grids in Table 2?

Authors admit that it was a bit confusing because the Tables 1 and 2 are not compatible in terms of the number of lines. In the revised manuscript, we correct this issue so that the shift of the center of mass does not appear to be confusing.

- Section 2.2.2 and further: I don't understand the meaning of "stratification" in this context.

This is also mentioned by other reviewers and we correct the terminology. It was meant to infer that the computed $L_p$ and DOLP differences are binned according to the along-track Laplacian (or that divided by L), but it was not the standard terminology. We remove this expression in the revised manuscript for clarity.

- As the cloud detection algorithm is new, it would be good to provide some numbers on false positives/negatives. What are the confidence levels for cloud (non-)detections?

The purpose of our provisional cloud mask is to separate pixels into two groups, i.e. pixels that are likely cloudy and the pixels that are likely clear-sky. Indeed it is ideal if we could provide the skill of the new cloud mask, but we believe that the development and evaluation of a cloud mask is not the scope of this paper. The criteria in Table 3 and 4 are not particularly new, and we listed all criteria so that the algorithm is well

described.

- l113 introduce all the acronyms.

Thank you for the comment. These are part of the product name that characterizes the type of the SGLI L1B products. As a name convention reference, we add a reference to the SGLI Data Users Handbook. In short, the first three letters indicate the subsystem of the SGLI instrument ("POL" for polarization, "VNR" for visible and near infrared, and "IRS" for infrared scanning subsystems), the fourth letter "D" indicates the observation mode ("D" for daytime data), and the last letter indicates the resolution ("K" for 1 km resolution, "L" for aggregated 1 km resolution).

- I suppose there is a valid reason why you select a power law of -5/3 (Kolmogorov)

The Kolmogorov power exponent -5/3 is used here because the previous studies on the cloud power spectrum use this value as a reference. Even though the original intent of the early papers were to associate the wind speed power spectrum to the marine stratocumulus cloud structure, later it was found that the value could be a representative value of spectrum power law slope even for other cloud types. We evaluate the impact of this assumption on the simulation in the discussion section.

- l145: Please provide a reference for the "observed empirical distribution function".

It might not have been clear, but this empirical distribution function is obtained from the analysis of the SGLI data as mentioned in the sentence in the same line.

- l150-158: Why not compute this in (Q,U) over which you can actually average?

Due to the flow of the paper it may not be very clear, but our averaging is performed over $X_{m60}$, $X_0$, and $X_{p60}$. The averaging over (Q,U) and the averaging over ($X_{m60}$, $X_0$, $X_{p60}$ ) produce the same results in the authors' understanding.

- l202: I don't understand "0.0010 (i.e. 2.2%)"

We greatly appreciate that the reviewer spotted this. The percent value is with respect

to the median DOLP, which should be clearly marked. We also found that the value is in a bin of $L_{AT}$ with small population, and the second digit was insignificant. Therefore, the value is corrected in the revised manuscript as follows: "about $1 \times 10^{-3}$ (i.e. 2% of median DOLP, 0.041)". The same correction applies to the part of the $L_p$

- It would be insightful to also present intensity and polarimetric images for typical simulated cloud scenes for particular power law distributions.

Thank you very much for the suggestion. As the simulation is performed for every 3MI pixel, the image size is 20×4, and not suitable for the visualization. It is possible to perform a simulation for a large scene only for the illustrative purposes, as shown in Figure 3. However, we are afraid that the inconsistent figure to the simulation method introduces more confusion than the understanding. The paper by Szczap et al. (2014, Geosci. Model Dev. 7. 1779-1801) is also helpful to visualize the impact of the methodology used in the simulation of cloud fields.

Again, the authors are very grateful to the reviewer for spending a significant time to go over this manuscript and carefully providing useful comments. We hope that we addressed completely the points raised by the reviewer, but if there are further questions or suggestions, we are always happy to cover those.

[Figure]

[Figure]

Fig. 1. Figure 1. The fraction of pixels within the POLDER specification (dark blue) and the 3MI specification (black) in each bin of along-track Laplacian. The density histograms of the along-track Laplacian

[Figure]

**Fig. 2.** Figure 2. The schematic diagram of the field-of-view by the (a) 3MI and (b) SGLI.. The black arrow shows the motion of satellite along the orbit, and the three position of the satellites along the tra

[Figure]

**Fig. 3.** Figure 3. Simulated clouds with the different power-law spectrum slope. (a) No correlation, (b) -5/3, and (c) -3.0.

---

## Author Comment (AC2) · 23 Dec 2020

Thank you very much for reading through our manuscript and providing us positive and constructive review. We are confident that the manuscript becomes more beneficial for the community by addressing to the points you raised. We are grateful for your inputs and we are more than happy to respond to your questions and comments:

- It would be better to use the term 'error' instead of 'noise'

We agree with the reviewer. The term "noise" may not be precise as it implies the completely random noise. In the revised manuscript, we replace the word "noise" with

"error".

- Also, I suggest to clearly quote values for bias and RMSE as measures of accuracy.

We understand the reviewer's suggestion, and we completely agree with him that the bias and RMSE are useful measures of the uncertainty, particularly when the error distribution is close to normal (or close to any distribution for which the population standard deviation exists). The problem for this particular case is that the distribution does not look like a normal distribution. We found that the histograms can be empirically fitted with t-distributions with degree of freedom between 2 and 4, which implies that the population variance likely exists, but providing the bias and RMSE might be misleading to the readers who are likely be using these parameters as a location and scale parameters of a normal distribution. In the attached Figs. 1 and 2, we show the histogram of the DOLP error and $L_p$ error together with the normal distributions constructed from the mean and sample standard deviation. For this reason, we don't find it significantly useful to provide the mean and RMSE in place of median and 9th-95th percentile range. If the reviewer has a strong recommendation and elaborated discussion, we would greatly appreciate that and we could consider again providing bias and RMSE in the revised manuscript.

- Probably here the 15.9-84.1 percentile range is more useful.

We agree with the reviewer that the addition of the 15.9-84.1 percentile range serve for the better characterization of the histogram. The numbers will be included in the revised manuscript.

- Please quote the bias and RMSE for all 4 categories in the abstract.

Providing specific numbers in the abstract would be useful. We add the median and 5th to 95th percentile range to the revised abstract as we consider they are less misleading parameter to characterize the distribution than the bias and RMSE.

- Which wavelength has been used for this study?

The wavelength used was 0.869 $\mu$m. It is briefly mentioned in the Section 2.2.1 (line 90), but in the revised manuscript, we emphasize this once again in Section 3 "Results".

- For clear-sky over land, I would expect the error at 860 nm is much larger than at 670 nm. I would recommend to show errors at both wavelengths.

We appreciate reviewer's insightful comments. We confirmed that the error is indeed smaller at 670 nm, and we add the results for the clear-sky over land in the revised manuscript.

- Please make clear that the resulting errors are on L1B product

Thank you very much for the encouragement. We emphasize that the resulting errors are on the L1B product in the revised manuscript.

- Please relate the errors to the total error budget of the 3MI.

The 3MI's target error in the polarimetry is defined on clear-sky homogeneous scene over dark surface (Fougnie et al. 2018 JQSRT), and the value is $5 \times 10^{-4}$ in terms of polarized reflectance. In Fig. 3, we show the fraction of pixels that satisfy the target accuracy for each bin of along-track Laplacian. Given that 68.2% of data falls within the $\pm 1\sigma$ (i.e. RMSE) if the error distribution follows the normal distribution, we could interpret that the specification is well satisfied in a particular bin when the fraction exceeds 68.2%. The figure shows that the 3MI target accuracy will be met for fairly homogeneous scenes (low along-track Laplacian), but with an increase of inhomogeneity, the target accuracy would be difficult to be met.

- How do these errors relate to errors on aerosol and cloud properties?

We are not aware of relevant study regarding the error in cloud retrievals, but the error estimation by the reviewer gives an insight into the accuracy needed for retrievals. The cloud droplet effective radius retrieval is unlikely be affected as it is sensitive to the geometric location of the rainbow, but the effective variance retrieval might be affected. In the revised manuscript we refer to the reviewer's work.

- Also the case of clear-sky over ocean would be interesting.

We add this class in the revised manuscript as in the histograms in Figs. 1 and 2. Indeed, the $L_p$ errors are small compared to the cloudy scenes. The DOLP errors are larger because of highly polarized dark ocean.

- Can the median be interpreted as the median of the $\Delta$DOLP distribution corresponding to a given value of $L_{AT}/L$?

Yes, exactly. We clarify this point in the revised manuscript.

- Why are these large parts in Fig. 7 (preprint) 'Gray hatched' but not in Fig. 6 (preprint)?

It was because Fig. 7 (preprint) took too wide range of abscissa compared to Fig. 6 (preprint). We adjust the right end of the abscissa to 95% of the data for both $L_p$ and DOLP errors in the revised manuscript.

- I would say this is not the 'prediction of noise/error' but rather an 'understanding of error'. I suggest to move this part (results from the Mote Carlo simulations) to an Appendix.

In the interpretation of the data, the reviewer's comment is reasonable and we agree that the part of Monte Carlo model may hamper the flow of the paper. Our intent by saying "prediction" is rather for unforeseen future missions or past missions without compatible orbit/swath polarimeter with higher spatial resolution. For those missions, the error estimates from the Monte Carlo model would be a "prediction", and we believe that it is important to deliver the message that such a prediction is proven valid at least at the spatial scale of 1 km (SGLI pixel size). We therefore move the method to the Appendix and summarize the result as a discussion in the revised manuscript.

Thank you very much again for your careful comments and constructive discussions. As mentioned above, nearly all your comments will be incorporated into the revised manuscript and we always welcome your questions and suggestions in the following

review process if deemed necessary.

**Fig. 1.** Histogram of degree of linear polarization error with median (magenta line), the normal distribution with mean and sample standard deviation of all data (orange), and the normal distribution with mean

[Figure]

**Fig. 2.** Histogram of polarized normalized radiance error with median (magenta line), the normal distribution with mean and sample standard deviation of all data (orange), and the normal distribution with mean

**Fig. 3.** The fraction of pixels within the POLDER specification (dark blue) and the 3MI specification (black) in each bin of along-track Laplacian. The density histograms of the along-track Laplacian is presen

---

## Author Comment (AC3) · 23 Dec 2020

Thank you very much for sparing your time to go through our manuscript. It is our pleasure to have you as a reviewer and we greatly appreciate your consistent attitude toward the constructive revision. All your inputs regarding the presentation issues are taken into account in the revised manuscript to the best of our understanding. But, if there is any remaining issue, we are happy to address them in the following review process. More detailed answers to the comments and questions are as follows:

1. I'd like to see a clearer description in the abstract and introduction that the co-registration and interpolation issue in 3MI (and POLDER) is due to the non-

[Figure]

simultaneous observation.

We appreciate that the reviewer pointed out the basic principle of the source of the error. The type of error that we are addressing in this paper is important for any polarimetric instrument that does not use the beam-splitter, as the error comes from the inevitable synthesis of temporally or spatially inhomogeneous data. The question is how to quantify them and how to mitigate them. The SGLI itself suffers from some interpolation errors, but way less than that of the 3MI and POLDER. In the revised manuscript, we bring this point at the very beginning of the abstract. We also insert a paragraph in an appropriate part of the introduction.

2. To be consistent with Povey and Grainger paper, you should use the term 'error'.

We agree with the reviewer that the term 'noise' is not precise. We replace the term 'noise' with 'error' in the revised manuscript.

3. Is there a reference for the meaning of the file format names that aren't spelled out?

Yes, we reference the SGLI Data Users Handbook in the revised manuscript. We also add a brief description.

4. I would think "classification" of data, not "stratification", is a more appropriate.

Thank you for the comment. We correct the terminology according to the comment in the revised manuscript.

5. The coastlines appears to be significantly biased but not clouds, particularly over land. Color scale issue?

Figure 1 is the same figure with a modified color scale. The error over the land cloud is not still evident, and our understanding is that the land at 0.869 $\mu$m over this region is bright enough that the reflectivity contrast to the cloud is not as significant as that over the ocean. In the revised manuscript, we include the figure with the new color scale.

6. How would cloud motion during the filter wheel acquisition period affect the results?

[Figure]

It is an interesting point to be considered for the co-registration of multiple views of the same cloud. Indeed, the Multi-angle Imaging Spectroradiometer (MISR) team considers the cloud motion during the co-registration. However, it is unlikely that it affects the polarimetric accuracy of the 3MI as the acquisition time interval is 0.25 seconds. If a cloud element travels for, say 1% of a 3MI pixel (0.04 km = 40 m), the wind speed must be 160 m/s, which is unrealistic except for very extreme locations.

7. The word "stratified" is used, but I think you mean "correlation".

The expressions are corrected in the revised manuscript as the reviewer encourages.

8. I missed the explanation of why you are dividing the Laplacian by L.

This is because we found from the preliminary study that the relation between $L_p$ error and the along-track Laplacian ($L_{AT}$) does not strongly depend on the value of L. As the DOLP is defined as the $L_p/L$, we expect that dividing the $L_{AT}$ by L would be helpful to sort out the DOLP error. This description is added at the end of Section 2.2.2.

9. I'm not sure the Discussion section is really any different than a part of the results.

In the revised manuscript, we move the Monte Carlo model results into the discussion so that it serves as the discussion on the understanding of the noise structure, rather as the primary results of the study. This responds to the Dr. Otto Hasekamp's comment, as well.

10. What would be nice thing to include in a discussion section is some thoughts about how these results can be used.

Thank you very much for the suggestion. We add a subsection that summarizes our thoughts about the possible application of the current results.

11. Are the shifting weights described in Table 1 and 2 the same for all view angles?

Yes, it is the same for all view angles.

12. It would be nice to see more discussion of how large these errors are compared to the overall 3MI uncertainty. Are they the main source of uncertainty? What is the significance of these uncertainties for the ability to retrieval geophysical parameters?

The 3MI's target error in the polarimetry is defined on clear-sky homogeneous scene over dark surface (Fougnie et al. 2018 JQSRT), and the value is $5 \times 10^{-4}$ in terms of polarized reflectance. The radiometric noise is anticipated to be less than this, and as Fig. 2 shows, the motion-induced error can contribute significantly to the total error budget. At this point, we cannot say for sure that it will be the dominant error source, but at least we could mention that the magnitude of estimated error is larger than the mission specification except for homogeneous scenes. In the revised manuscript, we add a dedicated paragraph to place the results into the context of the mission requirements. The retrieval accuracy depends highly on the specific details of the retrieval techniques, but we point to the Dr. Otto Hasekamp's study in the introduction. We are not aware of relevant study regarding the error in cloud retrievals.

We greatly appreciate your constructive comments that let us make the manuscript more accessible for readers. Once again, thank you very much for your substantial encouragements.

[Figure]

[Figure]

**Fig. 1.** (a) The DOLP difference between proxy and reference data. (b) The visible composite of the SGLI Level 1B data the same zone (visualized by authors, original data by JAXA).

[Figure]

**Fig. 2.** The fraction of pixels within the POLDER specification (dark blue) and the 3MI specification (black) in each bin of along-track Laplacian. The density histograms of the along-track Laplacian is on top

---

## Author Comment (AC4) · 23 Dec 2020

We discovered after the submission that the figure captions are cut because of length limit. This correction provides the complete figure captions for the reviewer's comment.

Figure 1. The fraction of pixels within the POLDER specification (dark blue) and the 3MI specification (black) in each bin of along-track Laplacian. The density histograms of the along-track Laplacian is presented on the top. The green horizontal line shows the fraction of 0.682, which corresponds to the fraction of data within $1\sigma$ of a normal distribution.

Figure 2. The schematic diagram of the field-of-view by the (a) 3MI and (b) SGLI.. The black arrow shows the motion of satellite along the orbit, and the three position of the satellites along the track are shown to highlight asynchronous acquisition by the 3MI sensor.

Figure 3. Simulated clouds with the different power-law spectrum slope. (a) No correlation, (b) -5/3, and (c) -3.0.

---

## Author Comment (AC5) · 23 Dec 2020

We discovered after the submission that the figure captions are cut because of length limit. This correction provides the complete figure captions for the reviewer's comment.

Figure 1. Histogram of degree of linear polarization error with median (magenta line), the normal distribution with mean and sample standard deviation of all data (orange), and the normal distribution with mean and sample standard deviation between 5th and 95th percentiles (green). The green bar on the bottom shows the range between 5th and 95th percentiles.

[Figure]

Figure 2. Histogram of polarized normalized radiance error with median (magenta line), the normal distribution with mean and sample standard deviation of all data (orange), and the normal distribution with mean and sample standard deviation between 5th and 95th percentiles (green). The green bar on the bottom shows the range between 5th and 95th percentiles.

Figure 3. The fraction of pixels within the POLDER specification (dark blue) and the 3MI specification (black) in each bin of along-track Laplacian. The density histograms of the along-track Laplacian is presented on the top. The green horizontal line shows the fraction of 0.682, which corresponds to the fraction of data within $1\sigma$ of a normal distribution.

---

## Author Comment (AC6) · 23 Dec 2020

We discovered after the submission that the figure captions are cut because of the length limit. This correction provides the complete figure captions for the reviewer's comment.

Figure 1. (a) The DOLP difference between proxy and reference data. (b) The visible composite of the SGLI Level 1B data the same zone (visualized by authors, original data by JAXA).

Figure 2. The fraction of pixels within the POLDER specification (dark blue) and the

[Figure]

3MI specification (black) in each bin of along-track Laplacian. The density histograms of the along-track Laplacian is presented on the top. The green horizontal line shows the fraction of 0.682, which corresponds to the fraction of data within $1\sigma$ of a normal distribution.